# Human oligodendrocyte progenitor cells mediate synapse elimination through TAM receptor activation

Asimenia Gkogka [1], Susmita Malwade [1], Marja Koskuvi[1], Sohvi Ohtonen[1], Ellinor Molnar[1], Raj Bose [2], Sandra Ceccatelli [2], Jari Koistinaho [3], Jari Tiihonen [3,4,5], Martin Schalling [6], Samudyata Samudyata [1,7] & Carl M. Sellgren [1,5,7] ✉

Oligodendrocyte progenitor cells (OPCs) have been implicated in synaptic remodelling in animal models, but the underlying mechanisms and their relevance to human brain development remain unclear. Here, we generate a human multi-lineage forebrain organoid model in which OPCs, together with microglia, form close contacts with synapses and spontaneously internalize synaptic material. Single-nucleus transcriptomic profiling with unbiased cell-cell communication analysis identifies the growth arrest-specific gene 6 (GAS6)-TYRO3, AXL, and MERTK (TAM) receptor axis as a key signalling pathway, with neurons and microglia expressing GAS6 and a subset of OPCs expressing AXL. Further, dose-dependent pharmacological inhibition of TAM receptors demonstrates the importance of AXL, and targeted reduction of AXL expression in OPCs impairs synaptic uptake. These findings reveal a role for GAS6-AXL signalling in driving synaptic internalisation by AXL+ OPCs during early human brain development.

Oligodendrocyte progenitor cells (OPCs) give rise to oligodendrocytes (OLs), which produce insulating sheaths of myelin around axons to facilitate efficient transmission of electrical impulses. In humans, OPC generation begins in early second trimester and progresses in several spatiotemporal waves toward birth[1–4]. OPCs then expand and differentiate into OLs that initiate a highly coordinated programme of myelination[5,6], while surplus myelin and OPCs undergo refinement through microglial phagocytosis[7–9]. However, a significant proportion of OPCs do not differentiate into OLs[10] and may have important roles in the brain beyond oligodendrogenesis[11–15]. Similar to microglia, recent work suggests that OPCs contribute to sculpting the developing visual system in zebrafish and mice through engulfment of excess synapses[16–18]. The ability of OPCs to engulf thalamocortical inputs also decreases after microglia depletion, indicating a close coordination between the two glial subtypes within the developing brain[16]. Nonetheless, mechanisms used by OPCs to remove developmentally primed synapses as well as the molecular basis for coordination between OPCs and microglia remain to be characterised.

Models based on human induced pluripotent stem cells (iPSCs) have confirmed the capacity of human-derived microglia to engulf synaptic structures[19,20]. Experimental systems using patient-derived microglia have also been instrumental in discovering mechanisms contributing to aberrant synapse refinement in various neurodevelopmental and neurodegenerative disorders, while highlighting

[1]Department of Physiology and Pharmacology, Karolinska Institutet, Stockholm, Sweden. [2]Department of Neuroscience, Karolinska Institutet, Stockholm, Sweden. [3]Neuroscience Center, HiLIFE, and Drug Research Program, Division of Pharmacology and Pharmacotherapy, University of Helsinki, Helsinki, Finland. [4]Department of Forensic Psychiatry, University of Eastern Finland, Niuvanniemi Hospital, Kuopio, Finland. [5]Center for Psychiatry Research, Department of Clinical Neuroscience, Karolinska Institutet and Stockholm Health Care Services, Stockholm County Council, Stockholm, Sweden. [6]Department of Molecular Medicine and Surgery, Karolinska Institutet and Center for Molecular Medicine, Karolinska University Hospital, Stockholm, Sweden. [7]These authors jointly supervised this work: Samudyata Samudyata, Carl M. Sellgren. ✉e-mail: carl.sellgren@ki.se

species differences regarding the contributing molecular machinery[21–24]. Several protocols have been developed to generate OPCs from human iPSCs in two-dimensional and three-dimensional models[25–29], but their phagocytic abilities have not been investigated, making it uncertain to what extent OPC-mediated synapse refinement is a feature of human brain development and, if so, what mechanisms govern this process. Furthermore, brain organoids containing maturing OL-lineage cells in the presence of yolk sac-derived microglia are still missing, limiting our understanding of how these cell types could interact in sculpting the developing neuronal circuits under physiological and disease contexts.

To address these questions, we first established a human multi-lineage forebrain organoid model that harboured OL-lineage cells as well as microglia. OPCs within these organoids displayed spontaneous uptake of synaptic material to a degree comparable to that of microglia. Utilising droplet-based single nucleus RNA sequencing (snRNA-seq) profiling, we then generated unbiased cell-cell communication networks focusing on the communication between neurons, OPCs, and microglia. This revealed interactions between GAS6, secreted by neurons and microglia, and the TAM receptor tyrosine kinases (TAM-RTKs) AXL and MERTK in OPCs as a major signalling pathway in certain OPC subpopulations. We confirmed protein expression of AXL, but not MERTK, in OPCs, and observed that OPCs engaged in internalising synaptic material displayed higher immunofluorescence intensity for AXL. GAS6 protein, whose transcripts were mainly produced by neurons and microglia, was also deposited on OPCs and correlated with the amount of internalised synaptic material in each OPC. To investigate whether TAM-RTK activation promoted the uptake of synaptic material in OPCs, we used live imaging to study the uptake of isolated synapses in human iPSC-derived OPCs. Utilising a potent small-molecule to inhibit the activation of TAM-RTKs in a dose-dependent manner, we observed decreased uptake of synaptosomes, as well as synaptic structures within organoids, in a dose range that implicated AXL-mediated signalling. Finally, we confirmed that specific knockdown of AXL in AXL+ OPCs resulted in impaired uptake of synaptic material.

## Results

### Multi-lineage human forebrain organoids contain OL-lineage cells and microglia

To investigate OPC-mediated elimination of synapses in a human developmental context, we generated a multi-lineage forebrain organoid model (iPSCs derived from $n = 5$ subjects) that harboured OL-lineage cells in proximity to developing neurons, astroglia, and microglia, thus mimicking the complex microenvironment of OPCs in the developing human brain. Briefly, iPSC-derived forebrain-patterned neural progenitor cells (NPCs) were co-cultured with primitive yolk sac macrophage progenitor cells (PMPs) capable of giving rise to microglia (days in vitro, DIV 0)[30]. The resulting 3D structures were then cultured with factors promoting OPC proliferation and differentiation[31], in addition to those needed for neuronal and microglial survival (Fig. 1a). Consistent with ganglionic eminence (GE) patterning in the presence of Sonic hedgehog (Shh) pathway agonists, ventral progenitors emerged at 43 DIV alongside native OPC populations and innately developing microglia (Fig. 1b). At 130 DIV, we observed post-mitotic neurons of both glutamatergic and GABAergic identities, along with astroglia (Fig. 1c). From 43 DIV to 130 DIV, cultures showed an increase in MBP+ area, indicative of OPC differentiation into OLs, with MBP+ processes observed in proximity to neurites, yet compact myelin was not observed (Fig. 1d). Furthermore, we observed abundant neuronal synapses organised primarily along the organoid periphery (Fig. 1e), while multielectrode array (MEA) measurements on 100 + DIV organoids showed spontaneous firing, indicating neuronal activity (Fig. 1f). To further profile the cellular diversity in the generated forebrain organoids, we performed snRNA-seq on freshly dissociated organoids

at 250 DIV (4 organoids per line, $n = 2$ subjects) and captured high-quality transcriptomic profiles of *10,645* nuclei (median of *3,624* genes per nucleus; Supplementary Fig. 1a–d). Reference mapping of obtained clusters with previously characterised datasets of primary human fetal brain and organoid models[3,31–35], combined with inspection of cluster-specific differentially expressed genes (Supplementary Data 1) and canonical cell type signatures, aided in identifying key neurodevelopmental cell types, including neurons, microglia, ependymal cells, radial glia, glioblast progenitors, and diverse OL-lineage cells spanning multiple maturation and proliferation states (Fig. 1g, h, and Supplementary Fig. 1e–g). Transcriptomic correlations to previously published primary human fetal brain datasets revealed regional similarities to the second-trimester fetal telencephalon (Supplementary Fig. 2a–c). Comparison to other brain region-specific organoid models indicated transcriptional overlap primarily with the forebrain-directed organoid model (Supplementary Fig. 2d), consistent with our protocol design. Importantly, the OL-lineage cluster comprised proliferating OPCs, OPC populations and relatively mature OLs (Fig. 1i), displaying high transcriptional similarity with primary second-trimester OL-lineage cells (Fig. 1j), and other organoid-derived OPC populations (Supplementary Fig. 2e, f). Notably, while the abundance of microglia was relatively low, a feature consistent with other brain organoid models[19,36,37] and not fully representative of in vivo microglial proportions, the microglia that emerged within the organoid model exhibited high transcriptional similarity to primary fetal microglia from the developing human forebrain and displayed a core microglial signature[32] (Supplementary Fig. 2g). To quantify neuronal composition within the organoids, we scored neuronal clusters using curated gene signatures derived from human fetal cortical development[38], which classified 71.2% of neurons as inhibitory and 24.8% as excitatory, with the remaining 4.0% unclassified (Supplementary Fig. 2h). These proportions reflect the expected bias toward ventral forebrain identities under GE patterning conditions, as also observed qualitatively by positive immunolabelling of vGLUT1 and GABA (Fig. 1c).

### Spontaneous uptake of synaptic structures in OPCs

Given the abundance of active synapses in the generated forebrain organoids, we investigated whether innately developing OPCs were capable of spontaneous uptake of synaptic material. High-resolution confocal imaging at 130 DIV revealed that both microglia and OPCs made contacts with neuronal synapses of glutamatergic (PSD-95 +) and GABAergic (GEPH +) identities (Fig. 2a–e). Three-dimensional volumetric reconstruction revealed OPCs demonstrating synaptic engulfment in varying degrees of internalisation, ranging from surface association to fully phagocytosed post-synaptic terminals within phagolysosomal compartments, similar to microglia (Fig. 2d, e). Given that the organoids were strongly enriched in inhibitory neurons under our GE patterning conditions (Supplementary Fig. 2h, Fig. 1c), quantitative analyses focused on GEPH+ post-synaptic terminals as the predominant synapse type. Remarkably, OPCs and microglia spontaneously internalised similar volumes of these post-synaptic terminals when normalised to their cellular volumes (0.0054 and 0.0046 total GEPH volume per cellular volume for microglia and OPCs, respectively; mean ± s.e.m.; Fig. 2f).

### GAS6 secreted from neurons and microglia communicate with TAM-RTKs in OPCs

To explore intercellular communication between OPCs and neurons, as well as between OPCs and microglia, we then inferred unbiased cell-cell communication networks via the expression of ligand-receptor pairs[39] utilising the generated snRNA-seq dataset. Consistent with previous findings from animal models[8,16,18,40–42], signalling pathways central to synapse organisation, neurite outgrowth, and OL differentiation were inferred both between OPCs and neurons, as well as between OPCs and microglia (Fig. 3a). Interestingly, among

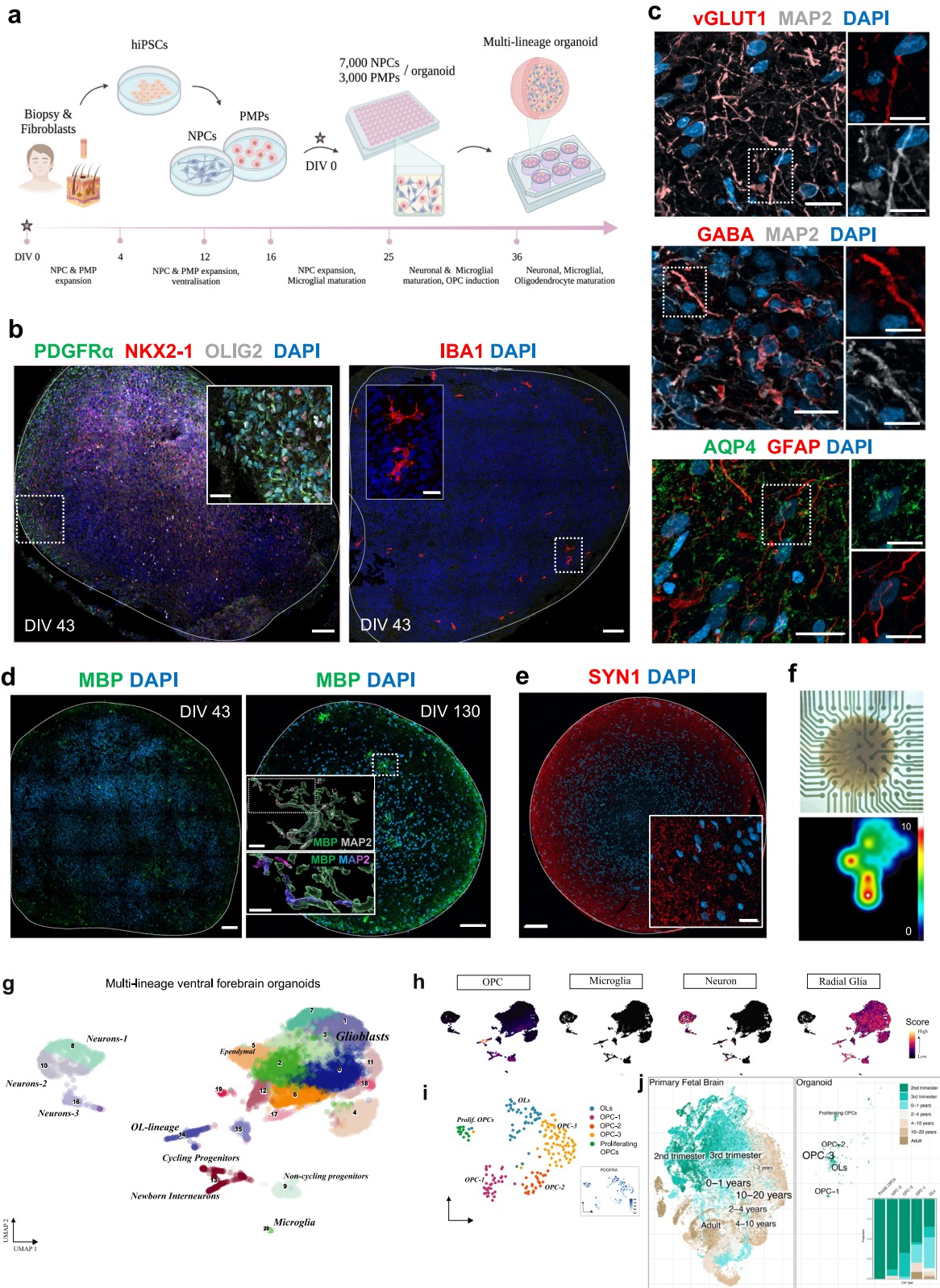

the top enriched signalling networks we also observed GAS6 (ligand) and TAM-RTKs (receptor) communication, which is a well-established pathway to mediate phagocytosis (Fig. 3a)[43–47]. GAS6 secreted by neurons (clusters Neurons-1 and Neurons-2), as well as from microglia, was inferred to activate the TAM-RTKs AXL and MERTK in OPCs (clusters OPC-1 and OPC-2; Fig. 3b, and Supplementary Fig. 3a, b).

## TAM-RTK activation by GAS6 correlates with ingestion of synaptic material in OPCs

TAM-RTKs (TYRO3, AXL, and MERTK) are known to mediate phagocytosis of diverse targets, such as apoptotic cells and myelin debris[44,45]. Their ligands, GAS6 and protein S1 (PROS1), bind to exposed phosphatidylserine (PtdSer) "eat-me" signals, triggering a conformational change that enables the ligands to robustly activate their receptor

**Fig. 1 | Multi-lineage forebrain organoids contain OL-lineage cells and microglia. a** Schematic summary of the protocol for generating multi-lineage organoids. Created in BioRender. Gkogka, A. (https://BioRender.com/fgiyqb1).
**b** Representative immunohistochemistry (IHC) images at 43 DIV showing ventral progenitors (NKX2-1, OLIG2) and OPCs (PDGFRα), in the presence of innately developing microglia (IBA1). **c** Representative IHC images at 130 DIV displaying mature neurons (MAP2) of glutamatergic (vGLUT1, top) and GABAergic (GABA, middle) identity, and astroglia (GFAP, AQP4, bottom). **d** IHC images of whole organoid sections showing myelinated areas (MBP) at 43 and 130 DIV, including an Imaris-based volumetric reconstruction showing an MBP+ cell wrapping MAP2+ neuronal processes with myelin (magnification, colour-coded by distance to MAP2: cyan = no contact, magenta = zero distance). **e** Representative IHC image of pre-synaptic elements (SYN1) across a whole organoid slice at 130 DIV. Representative images shown (**b**–**e**) were observed in at least two independent organoid batches derived from five iPSC lines with similar results. **f** Brightfield image of an intact multilineage organoid mounted on a 64-electrode MEA plate at 100 DIV (top), and corresponding activity heat map showing spike rate (spike/s) (bottom). **g** UMAP plot of *10,645* single-nucleus transcriptomes from forebrain organoids (250 DIV) across 20 clusters (numbers denote cluster IDs; each dot represents a cell coloured by cluster ID). Quality control for clustering displayed in Supplementary Fig. 1. Computed marker genes specific to each cluster ID provided in Supplementary Data 1. **h** UMAP plots displaying joint density distributions of cell type-specific gene signatures for OPCs[3], microglia[79], neurons, and radial glia (*HES1, HES6, SPARCL1, SOX6, DLX1, DLX5, EBF2, EOMES, NR2F2*; the colour bar represents joint kernel density). **i** UMAP plot of OL-lineage subclusters (cluster 14), with each dot representing a cell coloured by the respective subcluster. UMAP plot (in square) displaying the expression of PDGFRα in OL-lineage cells (colour scale representing expression kernel density). **j** Integrated UMAP of OL-lineage cells (cluster 14) from forebrain organoids (this study) and primary human OL-lineage cells (Velmeshev et al.[38]). Scale bars: 100 μm (**b**, **d**, **e**), 10 μm (magnification of **b**, **c**, top magnification of **d**, magnification of **e**), 5 μm (magnification of **c**, bottom magnification of **d**).

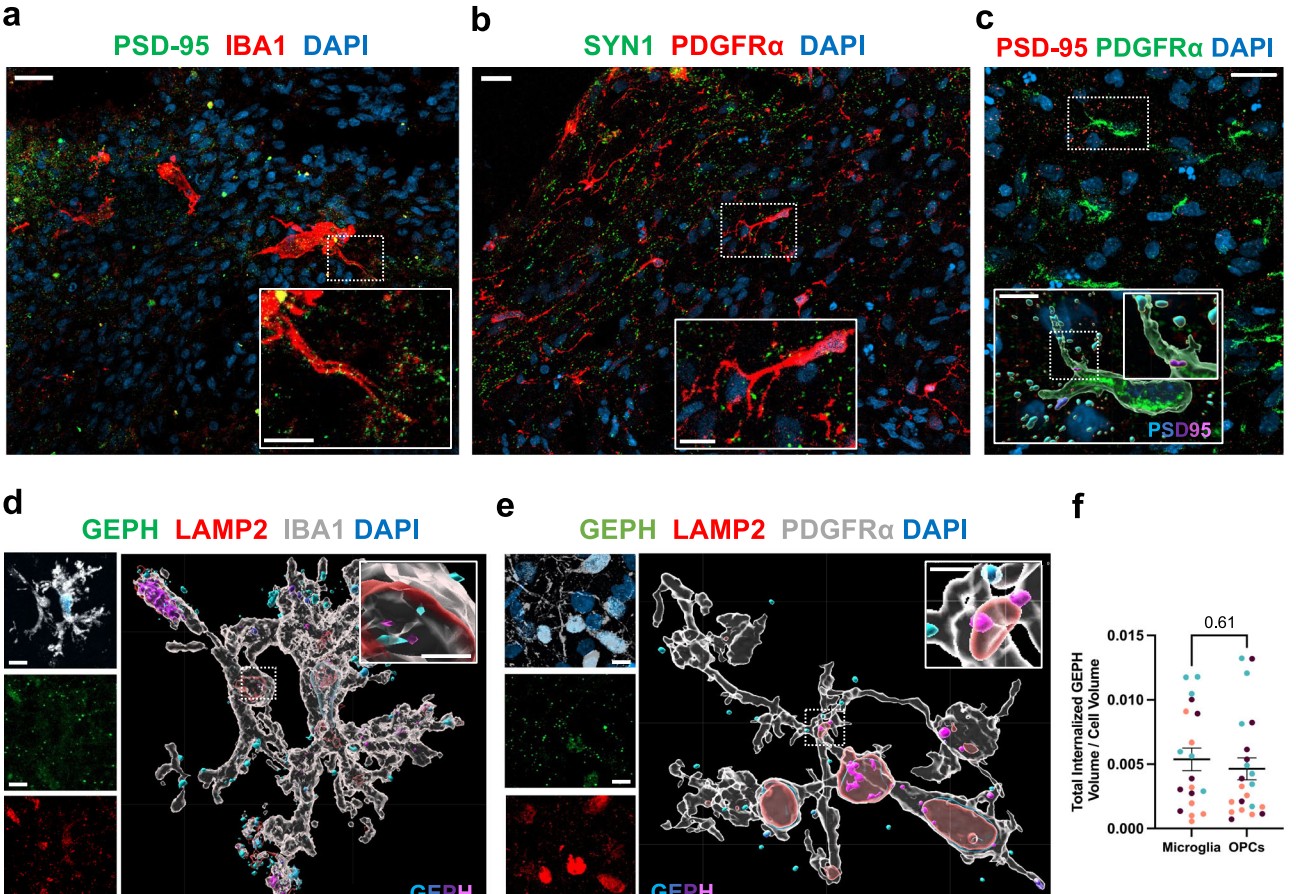

**Fig. 2 | Spontaneous uptake of synaptic structures by OPCs. a** Representative IHC image at 130 DIV showing microglia (IBA1) closely surveying synaptic terminals (PSD-95). **b** Representative IHC image at 130 DIV showing OPCs (PDGFRα) in extensive contacts with synaptic terminals (SYN1). **c** Representative IHC image at 130 DIV of OPCs displaying various levels of synapse internalisation, with an Imaris-based volumetric reconstruction (magnification) of an OPC engulfing post-synaptic excitatory elements (PSD-95). The statistics-based colour-code indicates the volumetric overlap of PSD-95 particles with PDGFRα+ surface (magenta = complete internalisation). Representative images (**a**–**c**) were observed in at least two independent organoid batches derived from five iPSC lines with similar results. **d** Imaris-based volumetric reconstruction of post-synaptic objects (GEPH) within microglia (IBA1), some localised within phagolysosomes (LAMP2). The statistics-based colour code indicates the volumetric overlap of GEPH puncta with IBA1+ surface (magenta = complete internalisation). **e** Imaris-based volumetric reconstruction of post-synaptic terminals (GEPH) within OPCs (PDGFRα), some localised within phagolysosomes (LAMP2). The statistics-based colour code indicates the volumetric overlap of GEPH puncta with PDGFRα+ surface (magenta = complete internalisation). **f** Quantification of total internalised volume of GEPH+ objects displaying ≥90% volumetric overlap with the surface of microglia or OPCs, normalised by respective cell volume. Data from 8 OPCs and 6 microglia per biological line (*n* = 3 lines indicated by colour). Bars show mean ± s.e.m. Median internalised volume: $3.89 \times 10^{-3}$ for MG, $3.36 \times 10^{-3}$ for OPCs. Two-tailed Mann–Whitney *U* test, *P* = 0.61. Scale bars: 20 μm (**a**–**e**), 10 μm (magnifications of **a**–**e**).

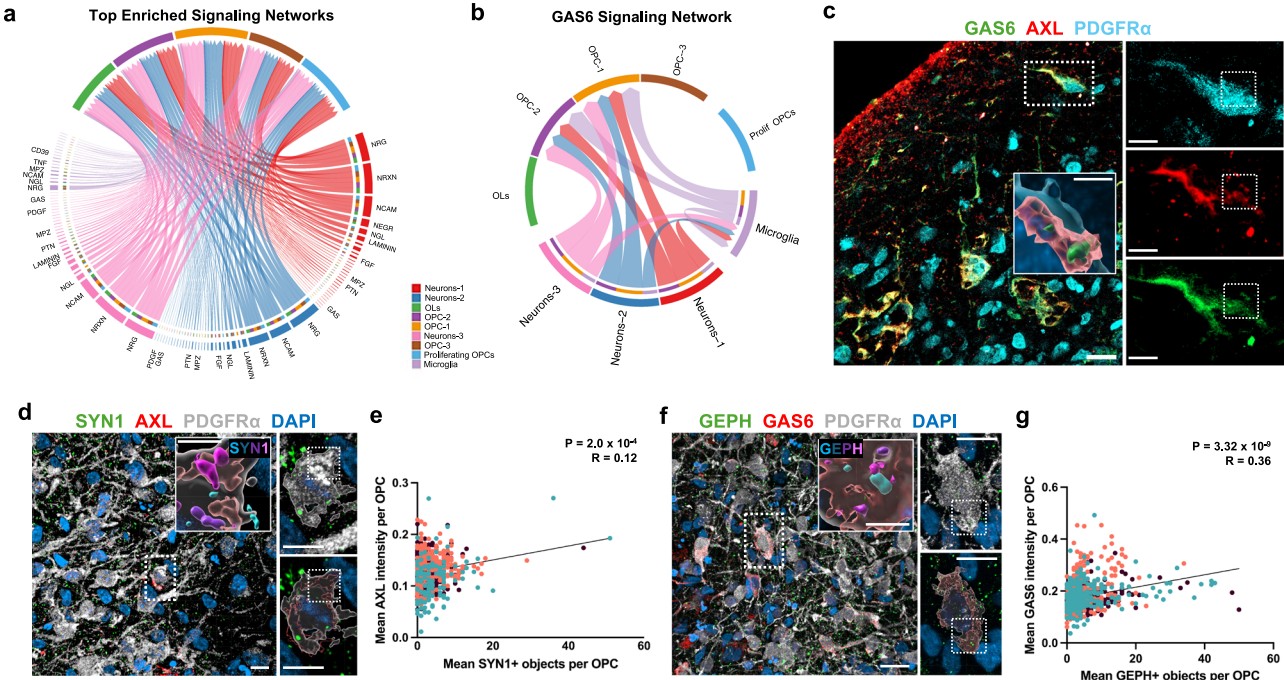

**Fig. 3 | OPCs interact with both neurons and microglia through GAS-TAM receptor signalling. a, b** Chord diagrams showing top enriched putative communication pathways (**a**) and interactions within the GAS6 signalling network (**b**) among OL-lineage cells, neurons, and microglia within the organoids, based on ligand-receptor expression analysis using CellChat[39]. Arrows indicate directionality of signalling; chord width represents the probability of communication for each ligand-receptor pair (Wilcox test, *P* < 0.05, one-sided). **c** Representative IHC image of OPCs (PDGFRα) in 130-DIV-old organoids, with Imaris-based volumetric reconstruction (magnification), showing co-localisation of AXL with its ligand GAS6 on the PDGFRα+ surface. **d** Representative IHC images of AXL-expressing OPCs (PDGFRα) in 130-DIV-old organoids, with Imaris-based volumetric reconstruction (magnification) showing synaptic uptake in OPCs. The statistics-based colour code indicates the overlap ratio of SYN1 puncta with PDGFRα+ surface (magenta = complete internalisation). **e** Scatter plot showing correlation between AXL

expression and the number of SYN1+ objects per OPC (*n* = 618 OPCs; three biological lines represented by colour). **f** Representative IHC images of GAS6-activated OPCs (PDGFRα) in 130-DIV-old organoids, with Imaris-based volumetric reconstruction (magnification) showing synaptic uptake in OPCs. The statistics-based colour code indicates the overlap ratio of GEPH puncta with PDGFRα+ surface (magenta = complete internalisation). **g** Scatter plot displaying correlation between GAS6 and the number of GEPH+ objects per OPC (*n* = 705 cells; three biological lines represented by colour). For **e, g** lines represent linear regression with 95% confidence interval. Statistical significance assessed using two-sided Spearman's rank correlation, with no multiple-comparison adjustment. Spearman's correlation coefficient (R) and *P* values are indicated. Representative images (**c, d, f**) were observed in at least two independent organoid batches derived from three iPSC lines with similar results. Scale bars: 20 µm (**c, f**), 10 µm (magnification of **c** (right), **d**), 5 µm (in-image magnification of **c, d, f**).

(GAS6: TYRO3, AXL, MERTK and PROS1: TYRO3, MERTK)[46-49]. In line with our inferred neuron-microglia signalling network in the forebrain organoids (Fig. 3a, b), previous in vivo studies in mice have shown that TAM receptors play key roles in synapse and cell debris clearance. Specifically, MERTK and AXL deficiency leads to reduced microglial engulfment of apoptotic cells during adult neurogenesis[50,51], and MERTK has been implicated in astrocyte-mediated synapse elimination during visual system development[52]. To investigate a potential role of TAM-RTK activation in developmental synapse refinement by OPCs, we examined TAM-RTK expression in OPCs within the forebrain organoid model. We found that 16% of OPCs harboured AXL protein (Fig. 3c, and Supplementary Fig. 3c), closely matching the 14% reported in primary fetal cortical tissue[32], whereas MERTK protein was not detected (Supplementary Fig. 3d). GAS6 protein was also present on AXL + PDGFRα + OPCs (Fig. 3c), whereas OPCs lacking AXL showed almost no GAS6 deposition (Supplementary Fig. 3e), while its corresponding mRNA was predominantly expressed by neurons and microglia (Supplementary Fig. 3f).

Further, AXL and GAS6 protein levels (quantified by immunofluorescence intensity) correlated with the number of synaptic puncta internalised by each OPC (Fig. 3d–g). Additionally, by leveraging a large-scale transcriptomic database of human brain tissue[53], we observed that co-expression networks centred on *AXL*, but not *MERTK*, exhibited a significant enrichment of synaptic genes (*P*-adjusted <0.05; Supplementary Fig. 3g, h, and Supplementary Data 2-3).

## AXL inhibition impairs the uptake of synaptic material in OPCs

To confirm a mechanistic role of TAM-RTK activation in OPC-mediated synapse elimination, we derived OPCs from iPSCs (*n* = 3 subjects) in monocultures (Fig. 4a, and Supplementary Fig. 3i) and exposed them to human, pHrodo-labelled synaptosomes derived from iPSC-derived neurons. Immunocytochemistry confirmed that these cultures were devoid of microglia (Supplementary Fig. 3i), consistent with the absence of mesodermal progenitors in the differentiation protocol. Although these OPC monocultures lack both neurons and microglia, immunostaining showed that synaptosomes–in addition to the synaptic markers (SYN1 and GEPH) – contained detectable levels of GAS6 (Fig. 4b), the ligand for AXL, suggesting that neuronal GAS6 on synaptosomes may be sufficient to engage AXL on OPCs. Live-imaging assays[20,24] also revealed robust internalisation of synaptic material in iPSC-derived OPCs (Fig. 4c, d), and subsequent fixation with volumetric reconstruction confirmed the intracellular localisation of pHrodo-labelled particles within PDGFRα + OLIG2+ OPCs, as well as the presence of the synaptic marker PSD-95 on synaptosomes (Fig. 4e). We further confirmed that these iPSC-derived OPCs express AXL and minimal levels of MERTK (Fig. 4f), a pattern consistent with that observed in organoid-grown OPCs, and used the potent small-molecule UNC2025 to inhibit phosphorylation and activation of TAM-RTKs in a dose-dependent manner[54]. Pre-treatment of AXL+ OPCs with UNC2025 (50, 150, or 200 nM for 1 h), followed by exposure to synaptosomes, led to a

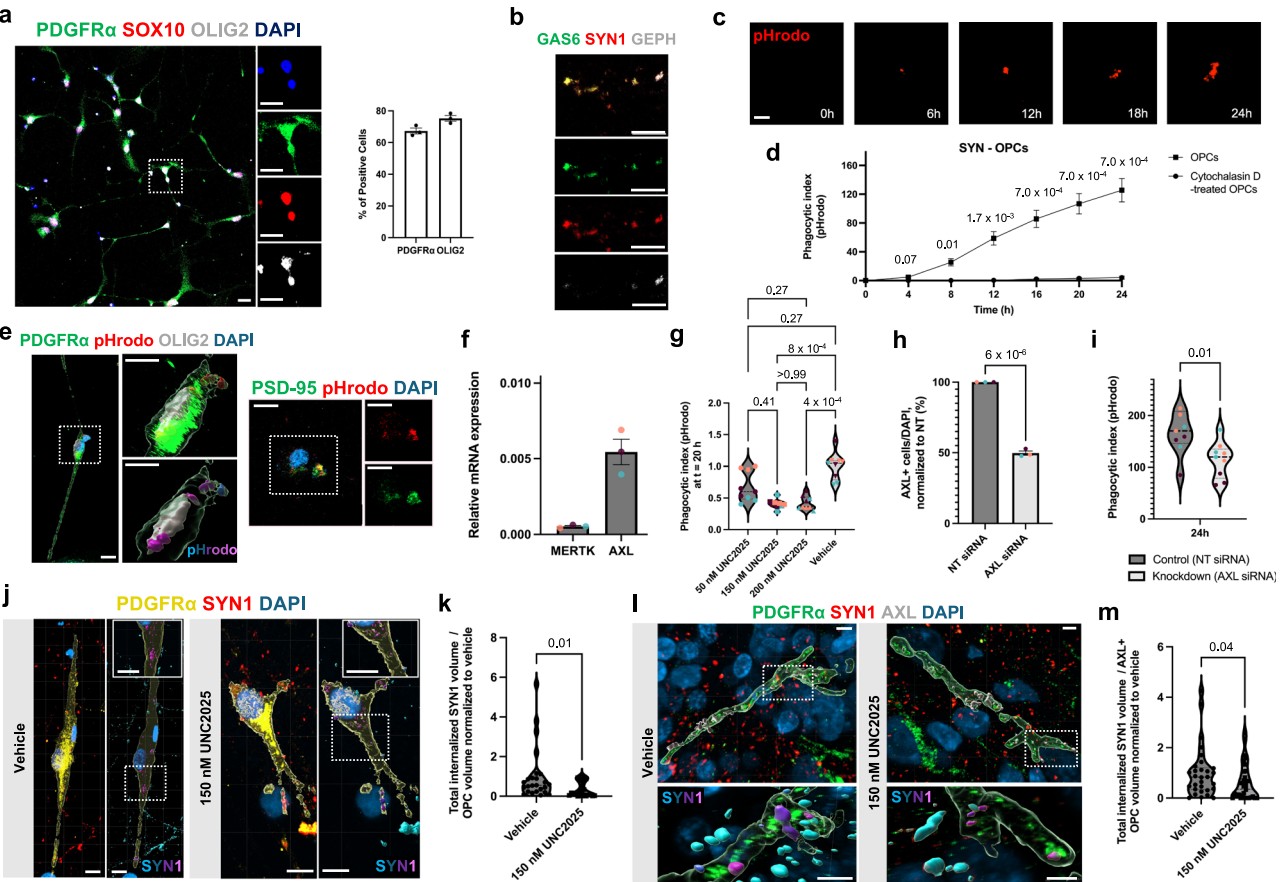

**Fig. 4 | AXL activation promotes uptake of synaptic material in OPCs.**
**a** Representative images of OPC monocultures displaying typical OPC morphology and high yield of PDGFRα + OLIG2+ cells (mean ± s.e.m, n = 3 biological lines). **b** Immunostaining of neuron-derived synaptosomes expressing the ligand GAS6 and synaptic markers SYN1 and GEPH. **c** Live-cell imaging (IncuCyte) showing internalisation of synaptosomes into acidic phagolysosomes of OPCs, indicated by red fluorescence from pHrodo. **d** Quantification of pHrodo-labelled synaptosome uptake in cytochalasin D (10uM)-treated and untreated OPCs (phagocytic index = mean pHrodo+ area per OPC; mean ± s.e.m, n = 3 lines; two-way repeated-measures ANOVA with Sidak's tests). **e** Confocal images and Imaris-based volumetric reconstructions confirming engulfment of pHrodo-labeled synaptosomes by OPCs (magenta = complete internalisation, left), and co-localisation with PSD-95+ puncta (right) indicating internalised synaptic material. **f** Relative mRNA levels of TAM receptors *MERTK* and *AXL* normalised to *GAPDH*. **g** Phagocytic indices in OPCs treated with 50, 150, or 200 nM UNC2025 or vehicle (DMSO) at 24 h (mean ± s.e.m., n = 3 lines represented by colour; Kruskal-Wallis with Dunn's post hoc).

**h** Quantification of AXL-expressing cells normalised to total nuclei (DAPI) showing -50% knockdown following *AXL* siRNA treatment versus non-targeting control (NT; unpaired two-sided t-test, no multiple-comparison adjustment). **i** Comparison of phagocytic indices in NT- versus *AXL* siRNA-treated OPCs at 24 h (median: 170.2 vs 19.5; two-tailed Mann-Whitney *U* test; n = 3 lines represented by colour). **j** Representative immunostaining and volumetric reconstructions of PDGFRα + OPCs showing SYN1+ internalisation in vehicle- and UNC2025-treated neuron-OPC co-cultures at day 26 (magenta = complete internalisation). **k** Quantification of total internalised SYN1+ volume overlapping ≥90% with OPC surface, normalised by cell volume, showing reduced uptake with 150 nM UNC2025 (median: 0.56 vs 0.17; n = 25 OPCs per condition). **l** Representative immunostaining and volumetric reconstructions of AXL+ OPCs showing SYN1+ internalisation in vehicle- and UNC2025-treated forebrain organoids. **m** Quantification of total internalised SYN1+ volume, normalised by cell volume, showing decreased uptake with 150 nM UNC2025 (medians: 0.83 vs 0.35; n = 30 cells per condition; two-tailed Mann-Whitney *U* test). Scale bars: 20 μm (**a**–**c**), 10 μm (**j**), 5 μm (**e**), 3 μm (**l**).

---

significant decrease in synaptosome uptake at both 150 nM and 200 nM concentrations after 24 h (Fig. 4g). As the decrease in uptake was similar at the two concentrations, and 150 nM is sufficient to completely inhibit AXL, while 200 nM is required to additionally inhibit TYRO3[54], this confirmed our findings from the organoid model, corroborating that AXL has a more prominent role than the other TAM-RTKs in mediating internalisation of synaptic elements by AXL+ OPCs. We then proceeded with inhibition of AXL using short interfering RNA (siRNA) targeted against *AXL* mRNA in OPCs. This RNAi-mediated knockdown reduced AXL protein levels by at least 50% (Fig. 4h, and Supplementary Fig. 3j) and resulted in a corresponding decrease in the uptake of pHrodo-labelled synaptosomes by 30% (P = 0.011; Fig. 4i, and Supplementary Fig. 3k), providing direct evidence for the involvement of AXL activation in promoting uptake of synaptic material in a subpopulation of human OPCs that express AXL.

To assess the functional role of AXL in synapse uptake by OPCs in a more physiologically relevant context, we next established co-cultures of iPSC-derived OPCs and neurons (Supplementary Fig. 3l) and treated the cultures with 150 nM UNC2025 or vehicle control between days 22-26. Immunostaining followed by 3D reconstruction revealed a reduction in the uptake of SYN1+ material by AXL+ OPCs upon UNC2025 treatment (Fig. 4j, k), confirming the role of AXL and TAM-RTK signalling in OPC-mediated synaptic uptake under co-culture conditions.

Finally, to avoid effects modulated by inhibition of AXL in microglia, we utilised ventral forebrain-patterned organoids lacking PMPs and exposed them to UNC2025. First, we confirmed that OPCs retained the ability to internalize synaptic material also in absence of microglia and then we observed a marked reduction in synapses internalised specifically by AXL+ OPCs in inhibitor- compared to vehicle-treated organoids (Fig. 4l, m), supporting the conclusion that AXL signalling promotes synaptic engulfment by AXL+ OPCs.

## Discussion

The refinement of developing neuronal circuits relies critically on the removal of excess synapses[55]. While it has been long known that microglia refine synapse numbers in the developing brain[20,21,56–59], more recent studies in mouse and zebrafish models have revealed that OPCs also engage in synaptic remodelling[16–18]. In microglia, signalling through complements factors[58,59] as well as through receptors such as TREM2[60] and GPR56[61], has been shown to promote internalisation of synaptic inputs. Despite these advances in our understanding of microglial uptake of synapses, the molecular machinery governing internalisation of synaptic inputs by OPCs has remained unexplored. Intriguingly, emerging evidence in animal models suggests a coordinated effort between microglia and OPCs at the synapse, with microglial depletion leading to a diminished ability of OPCs to ingest synapses[16], though the molecular underpinnings of this communication remain elusive.

Our study addresses these knowledge gaps by demonstrating that certain subpopulations of human OPCs in the developmental context can internalize synaptic structures depending on AXL activation. GAS6, the AXL ligand secreted by neurons and microglia, was deposited on AXL-expressing OPCs, indicating an underlying paracrine signalling capable of "tuning" and directing OPC-mediated synapse uptake. This aligns with animal studies showing a reduced ability of OPCs to engulf thalamocortical inputs following depletion of microglia, one of the two major sources of AXL-activating GAS6, though residual uptake persists, potentially mediated by neuronal-derived GAS6[16]. Consistent with this, we found that, even in the absence of microglia, OPCs retained the capacity to internalize synaptic elements, underscoring their intrinsic ability to engage in synaptic remodelling. Notably, organoid-grown OPCs spontaneously internalised synaptic material at levels comparable to microglia, mirroring recent in vivo observations in which OPCs and microglia exhibited comparable internalisation of thalamocortical inputs during critical periods of sensory-dependent refinement[16].

Although it remains unclear whether OPC-mediated internalisation of synaptic inputs is altered under pathophysiological conditions, the evolutionary expansion of synapse elimination in humans[62–65], combined with human-specific mechanisms, such as the specialisation of *C4* genes[24,66,67], may render these synaptic refinement processes vulnerable to defects. In line with this, several neurodevelopmental disorders have been linked to aberrant internalisation of synaptic inputs by microglia, while similar changes have been observed under neurodegenerative conditions, possibly through reactivation of neurodevelopmental programmes[68]. Notably, AXL upregulation has been reported in neurodegenerative and neuroinflammatory conditions characterised by synapse loss, such as Alzheimer's disease[43,69–71], traumatic brain injury[72], and ageing[71]. Further, GAS6 administration in animal models of multiple sclerosis has also been shown to promote remyelination at certain disease stages[73–75], indicating a dual, context-dependent role in maladaptive and adaptive responses.

In summary, we hereby use a versatile organoid model to demonstrate that human OPCs are capable of internalising synaptic structures and that this process depends on GAS6-mediated activation of AXL in a subpopulation of OPCs. OPCs were predicted to respond to GAS6 secreted by microglia, providing a conceivable explanation for the decreased uptake of synaptic structures in OPCs after microglia depletion. Future in vivo studies investigating the regional and temporal aspects of AXL activation in OPCs under physiological and pathological conditions will shed light on the role of this mechanism in the healthy and diseased brain.

## Methods

### Collection of biopsies, generation, and maintenance of human iPSCs

All experiments involving human iPSCs were conducted in accordance with all relevant ethical regulations and approved by the Regional Ethical Review Board in Stockholm, Sweden (IRB approval number: 2023-05308-01). Dermal biopsies were collected from healthy adult donors after written informed consent following institutional and national guidelines, and fibroblast cultures were established using standard enzymatic dissociation and expansion protocols[1]. All samples were tested for mycoplasma (negative). Fibroblasts were reprogrammed into iPSCs using either mRNA reprogramming as previously described (2 lines) or viral reprogramming using the CytoTune-iPS 2.0 Sendai Reprogramming Kit, according to the manufacturer's instructions (3 lines). iPSC lines (two females and three males) were expanded in mTeSR1 on Matrigel-coated plates and purified via magnetic cell sorting (MACS) with Anti-TRA-1-60 MicroBeads.

### Generation and maintenance of PMPs

Human primitive macrophage progenitors (PMPs) were generated from iPSCs in a Myb-independent manner, as previously reported[30,76], to recapitulate primitive haematopoiesis of tissue-resident macrophages, such as microglia, from yolk sac progenitors before the emergence of hematopoietic stem cells. Briefly, 10,000 iPSCs were cultured for 4 days into a V-bottom ultra-low attachment 96-well plate in pre-PMP embryoid body (EB) medium, which consisted of mTESR1 supplemented with bone morphogenetic protein 4 (BMP-4, 50 ng/ml), stem cell factor (SCF, 20 ng/ml), vascular endothelial growth factor (VEGF, 50 ng/ml) and Y-27632 rho-kinase inhibitor (10 µM). The resulting EBs were transferred into tissue-culture treated 6-well plates containing PMP medium, which consisted of X-VIVO15 supplemented with interleukin-3 (IL-3, 25 ng/ml), macrophage colony-stimulating factor (M-CSF, 100 ng/ml), penicillin-streptomycin (P/S, 1X), β-mercaptoethanol (100 µM). Four weeks later, PMPs appeared in suspension, which were harvested for organoid generation.

### Generation and maintenance of NPCs

Human neural progenitor cells (NPCs) were generated from iPSCs, as previously described[30] with minor modifications. Briefly, 4000 iPSCs per well were cultured for 1 week into a V-bottom ultra-low-attachment 96-well plate to aggregate into EBs using EB medium, consisting of Advanced DMEM/F12, N-2 (1X), B27 without vitamin A (1X), dorsomorphin (1 µM) and SB431542 (5 µM). Resulting EBs were plated on Matrigel-coated plates in a medium composed of Advanced DMEM/F12, N-2 (1X) and laminin 521 (1 µg/ml) and developed in the form of rosettes for 1 week. Rosettes were then manually isolated from surrounding cells and expanded for 1 additional week in NPC medium, consisting of a 1:1 mixture of Neurobasal and Advanced DMEM/F12, supplemented with N-2 (1X), B27 without vitamin A (1X), basic fibroblast growth factor (bFGF, 20 ng/ml), human leukaemia inhibitory factor (hLIF, 10 ng/ml), CHIR99021 (3 µM), SB431542 (2 µM) and Y-27632 rho-kinase inhibitor (10 µM).

### Generation and maintenance of multi-lineage organoids

Organoids were generated by co-culturing 7000 NPCs and 3000 PMPs derived from three iPSC lines per organoid, as described previously[30] with some modifications. The resulting EBs were maintained into a V-bottom ultra-low-attachment 96-well plate in a 1:1 mixture of NPC and PMP medium for the first 4 days in vitro (DIV). PMP medium was composed of X-VIVO 15, supplemented with IL-3 (25 ng/ml) and M-CSF (100 ng/ml). For the next 11 days (DIV 5–15), organoids were cultured with a mixture of 1:1 PMP and pre-OPC medium, and on DIV 8 transferred to ultra-low-attachment 6-well plates and kept on the orbital shaker at a speed of 80 rpm. Pre-OPC medium consisted of basal medium, supplemented with bFGF (20 ng/ml) and EGF (20 ng/ml). Basal medium comprised of 1:1 mixture of Neurobasal and Advanced DMEM/F12, GlutaMax (1X), Non-Essential Amino Acids (NEAA, 1X), human insulin (25 µg/ml), β-mercaptoethanol (100 µM), P/S (1X), N-2 supplement (1X), and B-27 supplement without vitamin A (1X). During DIV 16-24, organoids were cultured in pre-OPC medium,

supplemented with microglial maturation factors interleukin-34 (IL-34, 100 ng/ml) and granulocyte-macrophage colony-stimulating factor (GM-CSF, 10 ng/ml). Further, to pattern these organoids towards the ventral forebrain fate, Wnt pathway inhibitor IWP-2 (5 μM) was added to the medium during DIV 4-24 and the small molecule smoothened agonist (SAG, 1 μM) during DIV 12-24. From DIV 25 until DIV 35, they were cultured in OPC medium, consisting of basal medium with triiodo-L-thyronine (T3, 60 ng/ml), biotin (100 ng/ml), neurotrophin-3 (NT-3, 20 ng/ml), cyclic AMP (cAMP, 1 μM), hepatocyte growth factor (HGF, 5 ng/ml), insulin-like growth factor 1 (IGF-1, 10 ng/ml), platelet-derived growth factor (PDGF-AA, 10 ng/ml), supplemented with IL-34 (100 ng/ml), GM-CSF (10 ng/ml) along with the neuronal maturation factors brain-derived neurotrophic factor (BDNF, 20 ng/ml) and glial cell-derived neurotrophic factor (GDNF, 20 ng/ml). From DIV 36 onwards they were grown in OL medium consisting of 2:1:1 mixture of BrainPhys neuronal medium, Neurobasal and Advanced DMEM/F12, respectively, with GlutaMax (1X), NEAA (1X), human insulin (25 μg/ml), β-mercaptoethanol (100 μM), P/S (1X), N-2 (1X) and B-27 without vitamin A (1X), supplemented with T3 (60 ng/ml), biotin (100 ng/ml), cAMP (1 μM), ascorbic acid (20 μg/ml), IL-34 (100 ng/ml), GM-CSF (10 ng/ml), BDNF (20 ng/ml) and GDNF (20 ng/ml). Medium changes were performed daily until DIV 15, every other day until DIV 43, and every 3 days from DIV 44 onwards. A minimum of 10 organoids were generated per line.

### Generation and maintenance of microglia-depleted ventral forebrain organoids

To assess the role of microglia in synapse internalisation by OPCs, ventral forebrain organoids were generated without microglia by omitting PMPs and microglia-supportive cytokines (IL-3, IL-34, M-CSF, GM-CSF). Aggregates were formed from 10,000 iPSCs per EB seeded in U-bottom ultra-low attachment 96-wells. From DIV 1 to DIV 3, EBs were cultured in Essential 6 supplemented with dorsomorphin (2.5 μM) and SB431542 (10 μM). From DIV 4-6, IWP-2 (2.5 μM) was added to the medium to promote ventral forebrain patterning. Media was changed daily during DIV 1-6. On DIV 7, organoids were transferred to ultra-low attachment 6-well plates on an orbital shaker and cultured in medium consisting of basal medium (1:1 DMEM/F12:Neurobasal, GlutaMAX (1X), B-27 minus vitamin A (1X), N-2 (1X), NEAA (1X), β-mercaptoethanol (0.1 mM), insulin (25 μg/ml)) supplemented with EGF (20 ng/ml), FGF2 (20 ng/ml), and IWP-2 (2.5 μM). From DIV 12-26, SAG (0.1 μM) was added to the medium to reinforce ventral identity. From DIV 26-36, organoids were grown in OPC specification medium containing basal medium supplemented with NT-3 (20 ng/ml), BDNF (20 ng/ml), GDNF (20 ng/ml), cAMP (1 μM), T3 (60 ng/ml), biotin (100 ng/ml), HGF (5 ng/ml), IGF-1 (10 ng/ml), and PDGF-AA (10 ng/ml). From DIV 38-44, organoids were maintained in differentiation medium containing basal medium supplemented with BDNF, GDNF, cAMP, T3, biotin, and ascorbic acid, From DIV 45 onwards, maturation medium consisted of a 2:1:1 mix of BrainPhys, Neurobasal-A, and DMEM/F12, along with GlutaMAX, B-27 minus vitamin A, N-2, NEAA, insulin, β-mercaptoethanol, and the same small molecules and growth factors as above. Medium changes were performed every 2-3 days, with volume adjusted for organoid growth.

### Cryopreservation

Following fixation in 4% PFA for 45 min at RT, the organoids were washed twice in PBS for 5 min and transferred to 30% sucrose solution (w/vol) for overnight incubation at 4 °C. Upon removal of sucrose, organoids were embedded in optimal cutting temperature (OCT) compound and stored at −80 °C until used for cryosectioning. Organoid blocks were then allowed to equilibrate to sectioning temperature in the cryostat chamber for 30 min prior to sectioning, when 18-μm-thick sections of frozen organoid tissue were prepared using a Leica cryostat and stored at −20 °C until use. The optimal temperature

of the blade and the chamber was at −20 °C and multiple serial sections were collected from each organoid, allowing for exploration of several markers across different regions of the organoids.

### Immunohistochemistry

Cryosections were washed with PBS, followed by permeabilisation, and blocking in 3% BSA, 0.1% Triton X-100 in PBS for 1 h at RT. The sections were then washed twice in PBS and incubated overnight at 4 °C with primary antibodies diluted in blocking solution. Upon washing twice with PBS, the cryosections were incubated in a mixture of appropriate secondary antibodies diluted in blocking solution for 2 h at RT in the dark. Stained samples were washed twice with PBS, incubated with DAPI for 5 min and mounted on glass slides using Fluorescence Mounting Medium. For anti-PDGFRα, an initial permeabilisation step with 0.2% Triton X-100 in PBS for 10 min was added before blocking, and a quenching step with TrueBlack® Plus Lipofuscin Auto-fluorescence Quencher diluted in 70% ethanol for 30 s was added before mounting. Imaging was performed on the confocal laser-scanning microscopes Zeiss LSM800 and LSM980-Airy using ZEN2009 software. ImageJ was used for image processing and Cell-Profiler was used for cellular quantifications. For myelination around neuronal processes, z-stacked images (10 μm range, 1.0 μm step-size) were acquired with 40x magnification (C-Apochromat 40x/1.20 W) across the entire organoid section, maximum intensity projections were generated using ImageJ software and images were then stitched and analysed using Imaris software. For data presentation, the volumetric reconstruction of an MBP+ cell was statistically colour-coded, ranging from cyan = 0 (no internalisation) to magenta = 1 (complete internalisation), using the "shortest distance to surface" function with the MAP2+ neuronal processes. All antibody information is listed in Supplementary Data 4.

### Spontaneous engulfment quantification within multi-lineage organoids

Z-stack images (10 μm range, 0.5 μm step-size) of DIV 130 organoid sections stained for OPC and/or microglia markers were acquired with 40X magnification (C-Apochromat 40x/1.20 W) across the entire organoid section. Images were pre-processed using ImageJ, by applying the "Enhance contrast" command, so that 0.1% of pixels would be saturated, and the "Mean filter" with a pixel radius of 1.5 μm. Then, each cell of interest was cropped into its own image stack file, which was used in downstream analysis, where maximum intensity projections were generated. In Imaris software, volumetric reconstructions of the fluorescence images were created using the "Surface" module to reconstruct the cell (PDGFRα, IBA1), the synaptic elements (GEPH, SYN1, PSD-95) and the lysosomes (LAMP2) of interest using the fluorescence as a reference. The synaptic units were defined as the pre-synaptic element surfaces (SYN1) with lower than 0.2 μm shortest distance from the post-synaptic element surfaces (GEPH) by using the filtering command and by creating a new surface consisting of pre- and post-synaptic elements within this distance range. The synaptic particles were defined as internalised when ≥ 90% of their volume ratio overlapped with the defined cell surface. The volumes of the cell and the internalised surfaces were collected from the statistics tab in Imaris. The total internalised volume was then normalised to a cell's volume to represent the amount of engulfment by each cell. For data presentation, the synaptic elements and unit surfaces were statistically colour-coded, ranging from cyan = 0 (no internalisation) to magenta = 1 (complete internalisation), based on the overlapped volume ratio with the cell surface.

### Multi-electrode array (MEA) recording

6-well MEA plates were coated with 0.1% polyethyleneimine solution for 1 h, washed twice with water, and one organoid per well was placed on the electrode grid in 100 μl of media supplemented with laminin 521

(10 µg/ml) for 2 h to facilitate attachment. For the next 48 h, the organoid maturation medium was supplemented with laminin 521 (1 µg/ml), followed by media changes every 2–3 days. The recordings were performed using a Maestro MEA system with AxIS Software Spontaneous Neural Configuration. Spikes were detected with AxIS software using an adaptive threshold crossing set to 5.5 times the standard deviation of the estimated noise for each electrode.

### Dissociation of organoids and snRNA-seq
Four organoids per biological line ($n = 2$) were processed for single nuclei (sn) RNA sequencing at DIV 250. To isolate single nuclei from fresh organoid tissue samples, organoids were subjected to cold mechanical dissociation with glass pipettes to obtain a single-cell suspension. Dissociated cells were centrifuged at 500 g for 5 min at 4 °C. The cell pellet was washed and exposed to a freshly prepared lysis buffer (as demonstrated in protocol CG000366 RevD, 10x Genomics) followed by centrifugation at 500 g for 5 min at 4 °C. To remove cellular debris, the nuclei suspension was further subjected to an iodoxinol gradient (Optiprep; 25% and 29%) with centrifugation at 13500 g for 20 min at 4 °C. The resulting nuclei pellet was resuspended in diluted nuclei buffer and observed under a Leica brightfield microscope at 40X magnification to determine nuclei quality. The nuclei concentration was manually counted using trypan blue and a Neubauer counting chamber. Isolated nuclei were equally pooled and loaded onto the Chromium controller (10x Genomics) with a target recovery of 10000 cells. The assay was processed according to the Chromium Next GEM Single Cell 3' Reagent Kits v3.1(Dual Index) user guide (RevD). Upon GEM generation, barcoding, cleanup and cDNA amplification, a single barcoded library was constructed and sequenced on Illumina NovaSeq S1-100 (v1.5, 2 ×100 bp).

### snRNA-seq data processing and analyses
Raw sequencing data obtained as fastq files were processed using the cellranger (v6.1.2) pipeline with read alignment to the human GRCh38/hg38 reference genome. Barcodes and sorted bam files generated by cellranger were used for genotype demultiplexing of the two individual cell lines by souporcell package, with k = 2 and a common variants file (limited to ≥2% MAF) of GRCh38. Droplets identified as empty or containing ambient RNA were excluded from further analysis. Each cell was assigned to the respective individual cell line. UMI count matrices were processed in R using the Seurat package. All cells were further subjected to quality control metrics according to the number of transcripts (0000) and genes ( >200 and < 11,000) captured, the percentage of mitochondrial transcripts (5%). Genes expressed in less than 3 cells were excluded. Cells were scored for the cell cycle phase in Seurat and the difference between the S phase and G2M phase scores was calculated. Doublets were identified and filtered out using DoubletFinder (pN = 0.25, pK = 0.22). Stressed cells within the organoid were identified using the Gruffi package. Post quality control, gene expression per cell was normalised and scaled using SCTransform function (3000 features) to account for technical artifacts and the cell cycle difference was regressed. Batch effects were evaluated and cells originating from individual cell lines were normalised individually and integrated in Seurat using FindIntegrationAnchors and IntergateData functions. To reduce dimensionality, UMAP algorithm was applied to the integrated dataset with 30 dimensions in PCA space as input. Unsupervised graph-based clustering was performed in Seurat using K-nearest neighbors' algorithm and Louvain algorithm (at 1.0 and 1.4 resolution) to find unique cellular clusters.

### RNA velocity and cellular trajectories
BAM files generated by cellranger were used as input to Velocyto package to obtain the spliced and unspliced counts matrices for all genes. RNA velocity analysis was performed using the scVelo package in Python with pre-computed dimensionality reduction and clustering.

Using the integrated data, the first-order and second-order moments were computed (default settings) and the velocities were calculated using the likelihood-based dynamical model. The velocity graph was visualised on the UMAP embedding as streams. Using the dynamical model, cellular trajectories were inferred along with their initial and terminal states within the dataset using CellRank toolkit in Python. Fate probabilities for each cell and driver genes for the OL lineage were computed using the tr.lineage drivers function.

### Differential expression and cell type annotations
Differential gene expression across identified clusters was performed using the FindAllMarkers function in Seurat using the MAST test (MALAT1 and mitochondrial genes were excluded). Multiple-testing correction was performed using the Benjamini-Hochberg method. The cluster markers along with known canonical marker gene expressions were visualised on the UMAP embeddings. Annotations were finalised based on a combination of unsupervised markers, transcriptomic correlations, and post-integration label transfers from multiple publicly available snRNA-seq datasets. External datasets from Braun et al.[33], Cameron et al.[77], Nowakowski et al.[32], Marton et al.[31], and Amin et al.[34]. were directly downloaded either as pre-processed R objects or gene expression count matrices with available annotations from respective publications. For cluster comparisons, clusters were transcriptomically correlated (Pearson's correlation) to external reference datasets to aid cell type annotations using the clustifyr package. Further, the dataset from this study was integrated using the Harmony package with external organoid datasets used as a reference to perform classification and transfer cluster labels using the TransferData function onto the query data. To quantify the excitatory and inhibitory composition of neuronal clusters, cells within neuronal clusters were scored using gene signatures obtained from fetal cortical neurons as described by Velmeshev et al. (signature pattern = "burst")[38]. Signature scores were computed for excitatory and inhibitory lineages, and a delta threshold of 0.01 was applied to assign each cell as excitatory, inhibitory, or unclassified. Cell-type proportions were subsequently computed across all neuronal clusters to assess neuronal subtype composition within the organoid model.

### Cellular crosstalk within organoids
Cell-cell communication networks across identified cellular clusters were computed and visualised using the CellChat toolkit (v1.6.0). A CellChat object was created from the Seurat object with discrete clusters and signalling genes. The ligand-receptor pairs for humans were extracted from the CellChatDB database. Overexpressed ligand-receptor interactions were identified and the 'trimeans' method was used to calculate average gene expression per cell group. The communication probability of all ligand-receptor interactions linked to known signalling pathways from the database was computed and the aggregated weighted-directed network across cell types of interest was visualised as circle plots and chord diagrams. To identify global communication patterns, unsupervised hierarchical clustering (k = 4) of signalling pathways across cell groups was performed. Significant interactions between cell types were determined using permutation testing and significance was considered at $P < 0.05$.

### Co-expression networks
Publicly available RNA-seq data derived from primary adult brain tissue of healthy individuals was downloaded from the GTEx Portal (v7). The pre-processed and normalised count matrix was used to compute correlations between the seed gene i.e., genes encoding the TAM receptors, individually. $P$ values were corrected for multiple testing using FDR < 0.01. A weighted undirected co-expression network was constructed via adjacency matrices using the igraph R package with vertices representing genes and edges representing the correlation (positive-red, negative-blue). Strength and direction of the correlation

was measured using pearson's correlation coefficient (PCC). Enrichment of synapse-related genes (obtained from the SynGO Database[78]) within each TAM receptor's network was tested using Fisher's exact test ($P < 0.05$, one-sided).

## Generation and maintenance of OPCs

Human OPCs were generated from iPSCs, as previously described[25] with minor modifications. Briefly, following differentiation from iPSCs, NPCs were expanded for one week in Neural expansion medium, composed of a 1:1 mixture of Neurobasal and Advanced DMEM/F-12, N-2 (1X) and B27 without vitamin A (1X), supplemented with bFGF (20 ng/ml) and epidermal growth factor (EGF, 20 ng/ml). Using Anti-A2B5 MicroBeads, A2B5+ cells were then isolated and plated on a fibronectin (25 µg/ml)-coated dish and cultured in OPC-like medium, composed of a 1:1 mixture of Neurobasal and Advanced DMEM/F12, supplemented with N-2 (1X), B27 without vitamin A (1X), PDGF-AA (10 ng/ml), IGF-1 (10 ng/ml), Sonic hedgehog (SHH, 100 ng/ml), forskolin (5 µM), N-acetyl cysteine (60 µg/ml), bFGF (20 ng/ml) and Noggin (20 ng/ml) for 15 days.

## Quantitative PCR (qPCR)

To quantify *TYRO3*, *AXL*, or *MERTK* expression in iPSC-derived OPCs, total RNA was extracted from lysed cell pellets using the DirectZol RNA-Miniprep Kit, following the manufacturer's instructions, including DNase I treatment. Quality and concentration of extracted RNA was determined using a NanoDrop (Thermofischer Scientific). Complementary DNA was synthesised using the High-Capacity RNA-to-cDNA™ Kit in 20 µl reaction volume containing 1 µg of input RNA with the following thermal cycler conditions: 37 °C (60 min), 95 °C (5 min). cDNA was further diluted to 1:3 and used as template for PCR reactions. Samples were prepared for qPCR in technical triplicates in 15-µl reaction volumes using PowerTrack™ SYBR Green Master Mix. qPCR reactions were performed on an Applied Biosystems QuantStudio 6 Pro Real-Time PCR System using the standard cycling protocol. Expression fold changes were calculated according to the ΔΔCt method, normalising to the housekeeping genes *GAPDH* (glyceraldehyde-3-phosphate dehydrogenase) and *ACTB* (actin beta).

## Immunocytochemistry

Cells were fixed in 4% paraformaldehyde (PFA) for 15 min at room temperature (RT) and washed twice with phosphate buffered saline (PBS). Permeabilisation and blocking were performed for 30 min at RT in 3% bovine serum albumin (BSA), 0.1% Triton X-100 (VWR) in PBS. Cells were washed twice in PBS and incubated overnight at 4 °C with antibody solution composed of 3% BSA, 0.1% Triton X-100 in PBS and the desired primary antibodies. Subsequently, cells were washed twice in PBS and incubated with antibody solution and appropriate secondary antibodies for 1 h at RT in the dark. Stained samples were washed twice in PBS, incubated with DAPI (1:500) for 5 min, and mounted with Fluorescence Mounting Medium. Confocal laser-scanning microscopes Zeiss LSM800 and LSM980-Airy with ZEN2009 software were used for imaging. ImageJ was used for image processing and analysis. All antibody information is listed in Supplementary Data 4.

**Quantification of AXL and GAS6 levels and correlation with synaptic uptake** Z-stack confocal images of PDGFRα + OPCs immunolabeled for AXL or GAS6 together with the synaptic markers SYN1 or GEPH were acquired using identical imaging settings across all samples. PDGFRα+ cells were used exclusively, as they overlap ~97% with double-positive SOX10 + PDGFRα+ cells. Images were processed in CellProfiler. PDGFRα + OPCs were identified using the IdentifyPrimaryObjects module, and synaptic puncta were detected within the same z-stack using a fixed intensity threshold and size filter (≥ 2 pixels in diameter) to exclude background signal. The number of SYN1+ or GEPH+ puncta and the mean fluorescence intensity of AXL or GAS6

within the cell boundary were quantified for each OPC for 618 or 705 cells, respectively, across three independent lines.

## Purification and labelling of synaptosomes

Synaptosomes were isolated from NPC-derived stochastically differentiated neuronal cultures using Syn-PER Synaptic Protein Extraction Reagent, as described previously[20,24]. Briefly, neurons were lifted off the plate with the use of a cell scraper, and the sample was centrifuged at 1200 g for 10 min. The supernatant was collected and centrifuged at 15,000 g for 20 min, and synaptosomes were collected in the form of a pellet and resuspended in medium containing 10% DMSO to protect against freezing artifacts. For real-time live-cell assays, synaptosomes were thawed, resuspended in 0.1 M sodium carbonate buffer (pH 9.0) and labelled according to the manufacturer's instructions with an amine-reactive and pH-sensitive dye (pHrodo red). Labelled synaptosomes were sonicated for 20 min to avoid clumping and resuspended in sodium carbonate buffer before its addition to the cells.

## Synaptic vesicle endocytosis by 2D OPCs

The IncuCyte ZOOM live imaging system was used at 37 °C and 5% $CO_2$, as previously described[20,24]. Briefly, sonicated and pHrodo-labelled synaptosomes were added to OPCs seeded at a 13,000-cells-per-well density in 96-well imaging plates and five images per well were acquired with the 10X objective every 1 h for a total period of 24 h (3 technical replicates, 5 images per replicate for each line, 3 lines). Images were pre-processed with the IncuCyte ZOOM software (version 2016A) and exported as PNG files. The phagocytic index refers to the ratio of the total red fluorescence emitted per well divided by the total number of OPCs present in each well every hour. Live imaging was followed by fixation of cells with 4% PFA for 15 min at RT followed by washing twice with PBS for immunocytochemistry assays and confocal microscopy.

## Pharmacological inhibition of TAM receptors in OPCs

Before the initiation of the phagocytosis assay, OPCs were pre-treated for 1 h with DMSO (vehicle), 50, 150, or 200 nM UNC2025 diluted in OPC-like medium. Complete medium change was performed immediately before the assay, when 1.5 µg pHrodo-labelled synaptosomes were added in the inhibitor- or vehicle-supplemented media. All experiments using UNC2025 were performed with 13,000 OPCs per well using 96-well plates ($n = 3$ biological lines, triplicates).

## siRNA-mediated *AXL* knockdown in OPCs

The Accell Human AXL siRNA SMARTpool (E-003104-00-0005) and Accell Non-targeting Pool (D-001910-10-05) were purchased from Dharmacon. AXL+ OPCs were treated with non-targeting control Accell siRNA or *AXL*-targeted Accell SMARTPool siRNA according to the manufacturer's instructions. Briefly, reconstituted Accell SMARTPool siRNAs and Lipofectamine 2000 were diluted in OPC-like medium immediately before use and added to the cells at a final concentration of 1 µM and 0.4 µl/well, respectively. OPCs were treated with siRNA for 10 days, at which time the synaptic vesicle endocytosis assay was performed using 1.5 µg pHrodo-labelled SYNs per well, followed by fixation in 4% PFA and immunocytochemistry for protein knockdown confirmation. All experiments using siRNA were performed with 13,000 OPCs per well in triplicates using 96-well plates ($n = 3$ biological lines, triplicates).

## Co-culture of NGN2-induced neurons and OPCs

Human iPSC-derived neurons were generated by doxycycline-inducible expression of NGN2, as previously described with minor modifications[24]. Briefly, iPSCs were plated at a density of 30,000 cells/well in 24-well plates coated with poly-L-ornithine (PLO, 20 µg/ml, overnight at 37 °C) followed by laminin (10 µg/ml, 2 h at 37 °C). Cells were cultured in neuron induction medium (NIM), consisting of

DMEM/F12 supplemented with N-2 (1X), NEAA (1X), GlutaMAX (1X), doxycycline (2 µg/ml), and Y-27632 rho-kinase inhibitor (10 µM, day 0 only). Media was changed daily for 3 days. On DIV 4, medium was replaced with neuron maturation medium (NMM #1), composed of DMEM/F12 and BrainPhys (1:1), supplemented with N21-MAX (1X), GDNF (10 ng/ml), BDNF (10 ng/ml), NT-3 (10 ng/ml), doxycycline (2 µg/ml), Ara-C (1 µM), and Cultrex laminin (1 µg/ml). A second media change with NMM #1 was performed on DIV 7. Neurons were maintained in NMM #1 until co-culture initiation. On day 11, human iPSC-derived OPCs were added to neuronal cultures at a density of 20,000 cells/well (24-well format) in NMM #2, a BrainPhys-based medium supplemented with N-2 (1X), B27 without vitamin A (1X), GDNF (10 ng/ml), BDNF (10 ng/ml), NT-3 (10 ng/ml), PDGF-AA (10 ng/ml), IGF-1 (10 ng/ml), forskolin (5 µM), Y-27632 (10 µM), and Cultrex laminin (1 µg/ml). At this stage, doxycycline was withdrawn, and astrocyte-conditioned medium (ACM) was added at 10%, increasing to 30% from DIV 14 onward. Half-media changes were performed every 2-3 days until fixation. Cells were fixed on DIV 26 for immunocytochemistry. Inhibitor treatments were applied during the final 5 days of co-culture (DIV 22-26) as indicated below.

### Pharmacological TAM receptor inhibition in co-cultures and organoids

Neuron-OPC co-cultures (at DIV 22-26) and non-microglia organoids (at DIV 97-104) were treated daily with either DMSO (vehicle) or 150 nM UNC2025 in their respective maintenance media. Media was replaced with freshly prepared inhibitor- or vehicle-containing media each day during the specified time windows. At the end of the treatment period, co-cultures were fixed immediately, and organoids were fixed and cryopreserved for downstream analysis of synaptic engulfment by AXL+ OPCs.

### Reporting summary

Further information on research design is available in the Nature Portfolio Reporting Summary linked to this article.

## Data availability

Single-nucleus RNA sequencing analysis and visualisation were performed in R (v4.2.2) and Python (v3.7.12) on macOS (Ventura 13.4). The sequencing data generated in this study have been deposited in the GEO database under the accession number GSE242275. Previously published datasets used for reference mapping and comparative analyses were obtained from Braun et al. (https://ega-archive.org/datasets/EGAD00001006049), accession number EGAS00001004107], Cameron et al. (https://figshare.com/articles/dataset/11629311), accession number EGAS00001006537], Nowakowski et al. (https://www.ncbi.nlm.nih.gov/projects/gap/cgi-bin/study.cgi?study_id=phs000989.v6.p1), accession number phs000989.v3], Marton et al. (https://www.ncbi.nlm.nih.gov/geo/query/acc.cgi?acc=GSE115011), accession number GSE115011], and Amin et al. (https://www.ncbi.nlm.nih.gov/geo/query/acc.cgi?acc=GSE233574), accession number GSE233574]. All microscopy- and qPCR-based quantifications, for which GraphPad Prism (v8.0) was used, are provided within the paper and its Source Data file. Source data are provided with this paper.

## Code availability

All custom code used for the single-nucleus RNA sequencing analysis is available at Zenodo under (https://doi.org/10.5281/zenodo.17418028) and in the GitHub repository at (https://github.com/SellgrenLab/organoid-oligodendrocyte).

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

## Acknowledgements

We thank the study participants and acknowledge the technical support provided by the Biomedicum Imaging Core (BIC) facility at Karolinska Institutet. The computations/data handling was enabled by resources provided by provided by the National Academic Infrastructure for Supercomputing in Sweden (NAISS) and the Swedish National Infrastructure for Computing (SNIC) at UPPMAX, Uppsala University. We are also grateful to Julschen Majkowitz for the valuable feedback on the manuscript. This work was generously supported by grants from Erling Persson Foundation (C.M.S), Hjärnfonden (C.M.S.: FO2022-0135), the regional agreement on medical training and clinical research between Stockholm County Council (C.M.S.: 2017-02559), Karolinska Institutet (C.M.S.: KID), the Swedish Research Council (S.C. 10815-20-4, the Strategic research area neuroscience StratNeuro (S.C), Karolinska Hospital ALF Research Support (M.S: FoUI-962690), the Swedish Research Council (M.S: 2022-02670), and the Swedish Brain Foundation (M.S: 2022-0357). The scheme in Fig. 1a was created in BioRender. Gkogka, A. (2025) (https://BioRender.com/fgiyqb1).

## Author contributions

S.S. and C.M.S. conceived the study and A.G. designed the experiments. A.G. generated microglia-containing organoids, with help from S.S. and M.K. S.O. generated microglia-depleted organoids. A.G. generated OPCs, performed and analysed synaptic engulfment assays, pharmacological and siRNA experiments, with help from S.S. A.G. generated neuron-OPC co-cultures, with help from E.M., and analysed their pharmacological treatment together with S.S. S.S., S.M. performed dissociations and prepared libraries for sequencing with help from A.G. S.M. analysed the transcriptomic data. A.G. performed immunostaining experiments, imaging and analysis, with help from S.S. MEA experiments were performed by A.G. and S.S. in collaboration with R.B. and S.C. M.S. contributed with methodological expertise and supervision. J.K. and J.T. contributed with collection of skin biopsies, reprogramming, karyotyping, and phenotypic data collection. A.G., S.S. and C.M.S. interpreted the data and wrote the manuscript with input from all co-authors.

## Funding

## Competing interests

The authors declare no competing interests.
