## [Transparent Peer Review file · Nature Communications]

Human oligodendrocyte progenitor cells mediate synapse elimination through TAM receptor activation

Corresponding Author: Professor Carl Sellgren

Version 0:

Reviewer comments:

Reviewer #1

(Remarks to the Author)

In this manuscript, the authors describe the role of GAS6-AXL signaling in promoting phagocytosis of neuronal synaptic terminals using a multi-lineage organoid system that includes microglia and OPCs generated from hiPSCs. They replicate the observations made by others using the rodent system and show that OPCs phagocytose synaptic material. They then performed an analysis of inferred cell-cell communication networks from snRNA-seq and identified GAS6-AXL as the ligand receptor pair likely to be involved in this process. This was verified by showing the presence of AXL mRNA levels in OPC monocultures and by immunodetection of GAS6 deposited on OPCs (mRNA made by neurons and microglia). The levels of AXL and GAS6 in OPCs correlated with the number of synaptic puncta internalized. To further establish the causal relationship, the authors inhibited GAS6-AXL signaling in OPC monocultures by a small molecule AXL inhibitor or AXL siRNA and showed that these manipulations reduced phagocytosis of human synaptosomes, as shown by a reduction in pHrodo+ phagocytotic index.

This is a timely study providing a novel mechanistic insight into the signaling pathways that are involved when OPCs internalize neuronal synaptic terminals. The manuscript is put together very tightly in four sets of figures, each containing to significant new data and organized in a lean and logical manner. The authors used an elegant multi-lineage organoid system to include both OPCs and microglia, as previous rodent studies had shown the requirement of microglia in OPC-mediated phagocytosis, though the authors did not show the role of microglia in their current human organoid culture model other than to show that the total internalized GEPH puncta / cell was similar between OPC and microglia.

I have a few minor points that could be addressed for improved clarification.

1) The results section describing spontaneous uptake of synaptic structures by OPCs (Figure 2) seems somewhat truncated and cryptic. While I realize that there is a page/word count limit, this part could be explained in more depth, as it describes the assay that forms the basis for all the key findings in the manuscript. The authors provide images for PSD-95 and SYN1 but perform 3D volumetric reconstruction using GEPH. The authors should explain their unique culture model and the cellular composition in more detail. Below are some specific points that should be addressed.

- a) What was the abundance of excitatory and inhibitory neurons? (They mention GE patterning but also show excitatory synapses).
- b) How are microglia involved in OPC-mediated phagocytosis of synaptic terminals? For example,
 - i) Are microglia always found near the sites of internalization of synapses by OPCs (i.e. at AXL+ sites?)
 - ii) Have the authors tried removing microglia from the multi-lineage organoids?
 - iii) If pHrodo signal increases in OPC monocultures in the absence of microglia (Figure 4), how is this finding interpreted? Are microglia not necessary for phagocytosis of synapses by OPCs? Are there a sufficient number of microglia among the 35% of the non-OPC cells in the OPC monocultures? Or does exposure to synaptosomes trigger a different signaling pathway?

2) Of the GAS6 ligands, TYRO3 might also be expressed in oligodendrocyte lineage cells (Zhang et al., 2014; Marques et al., 2018). Can GAS6 bind to TYRO3, and would GAS6-TYRO3 interaction influence/modulate OPC-mediated synapse uptake?

3) To what extent is GAS6 colocalized with AXL on OPC membranes? i.e. Could the authors show co-localization of GAS6

deposited by neuron/microglia and AXL endogenously produced by OPCs? Does the extent of GAS6-AXL colocalization along the the OPC surface correlate with the extent of synapse phagocytosis, perhaps to an even greater extent than the correlation with the level of AXL?

4) The data in Figure 4 show that AXL is involved in phagocytosis of exogenous human synaptosomes. Since these 'OPC monocultures' do not contain neurons or microglia, how is AXL activated in this culture system? Does the synaptosome prep contain GAS6 or does that bypass the need to have GAS6 activate TAM-RTKs on OPCs and somehow activates AXL on OPCs by a separate mechanism? How do the levels of AXL in OPC monocultures compare with those in the organoids that contain microglia and neurons?

Overall, this is a well written manuscript that effectively communicates exciting new data on the signaling pathway that mediates phagocytosis of neuronal synapses by OPCs in a human organoid culture system.

(Remarks on code availability)

Looks OK but I am not an expert in single nuclei RNA-seq bioinformatics.

Reviewer #2

(Remarks to the Author)

In this report, Gkogka et al aim to better characterize how oligodendrocyte precursor cells (OPCs) and microglia contribute to synaptic development, namely, by utilizing induced pluripotent stem cell (iPSC) and organoid based models. The authors employ use of IHC, scRNA-seq, and functional assays to identify that OPCs and microglia internalize synaptic elements within organoid models. Even further, the authors utilize ligand-receptor analysis tools to implicate neuronal and microglial GAS6-mediated activation of OPC internalization of synaptic elements. Live-imaging functional assays of engulfment using monocultured OPCs further demonstrate that inhibition of the GAS-TAM signaling pathway reduces OPC internalization of synaptic elements. Altogether, the authors' work helps further understanding of signaling events relevant to synaptic development, health, and maintenance within a novel, reductionist model. These findings also establish some foundation for future work to evaluate the GAS-TAM pathway as it relates to neurodevelopment, aging, and neurodegeneration. Some major notes are identified that relate to the authors' generalization of findings that require attention, with particular emphasis on better characterization and benchmarking of the oligodendrocyte lineage in their in vitro culture system. Minor notes relate to clarifying small sections that contain suspected error.

Major

1. The multilineage model is intriguing, but there are VERY few microglia (Fig 1b, 1g) and so it is important I think to be much more outward about the fact that the myeloid compartment is not truly reflective of in vivo proportions.
2. I have a general concern about the use of PDGF α (and even Olig2) in forebrain organoids to specifically mark OPCs. In this model and at these developmental stages, PDGF α can also mark glial progenitors (not necessarily OPC-specific). Therefore, a lot of the conclusions that depend on the PDGF α lineage may mistakenly be attributed to OPCs. I think additional stains like SOX10 would be much more convincing that cells of interest (e.g. Fig 3C) are truly in an oligo-committed lineage.
3. The single nucleus data in the UMAP in 1g is a bit confusing. At day 250 it is surprising that the OL-lineage appears to remain so immature in these cultures. Along these lines, I can understand the comparison to Marton et al., but I'd instead recommend benchmarking the oligos in this model with a primary fetal dataset and not another organoid dataset exclusively. Altogether, this analysis could be performed more robustly and could be presented more clearly to indicate the degree of OL-lineage differentiation present in this model. If it's true that the oligodendrocytes do not fully mature by day 250, this should also be discussed more clearly.
4. Lines 416-427, Figure 2: GEPH is used as a marker to measure microglial and OPC phagocytic capacity. GEPH is typically associated with inhibitory elements, and my concern is, that without additional markers to measure phagocytosis, the authors' conclusions may not be easily generalized all phagocytic functions. Can the authors provide data for any additional markers, namely for excitatory postsynaptic elements (PSD-95, for example)?
5. Lines 462-86, Figure 4: Similar to Major note (4.), the functional experiments use PSD-95 as a marker for phagocytosis, yet lack additional data on GEPH markers which were indicative of function in previous experiments. Altogether, consistency in markers and methods would benefit these works in order to strengthen the generalization of these findings.

Minor

1. Line 200: Specify "couple of hours" to a standard numeric value.
2. Lines 447-448: Reference to figure 3b-d seems mismatched, perhaps this is meant to be Fig. 3a-b or S3a-b?
3. Lines 447-450: Difficult to read through and would benefit from clarifying separation between the authors findings/models with that of others. See Minor note (2.) For additional notes of confusion for this section.
4. Line 462: Consider changing heading based upon experimental design. AXL was not activated in any way; it was found that increased inhibition of AXL reduces internalization by OPCs. Perhaps something like "AXL inhibition impairs uptake of synaptic material in OPCs" to be better reflect the experimental design and results. Otherwise, I would also recommend overactivation of TAM signaling via GAS6 supplementation, microglial/neuronal conditioned media, or genetic manipulations to further evaluate the question of TAM activation by GAS6.
5. I don't believe the velocity in the UMAP in 1g is necessary—if anything it is a bit distracting from interpreting the clusters.

(Remarks on code availability)

Reviewer #3

(Remarks to the Author)

In their manuscript entitled „Human oligodendrocyte progenitor cells mediate synapse elimination through TAM receptor activation” Gkogka et al. use a human iPSC-derived organoid system to assess the role of oligodendrocyte progenitor cells (OPCs) in synapse pruning. The authors suggest that similar to microglia, OPCs engulf synapses during development, which seems to be mediated via the AXL receptor – a phagocytosis receptor that is well known for playing an important role in microglia.

These results are quite interesting and suggest a yet unknown function of OPCs and is this of high importance for the field. Although the paper is quite good as a beginning, there are still fundamental experiments lacking before it can be published.

- Is there evidence that the expression of AXL is not only an in-vitro artefact? Some human (published data on this). Could it be that it is only transient during development (as represented by the organoid model? This suggestion is a yet unknown function of OPCs and thus of really important. Can this be seen in mouse models during different stages of development and in the adult brain? Or is it human-specific?
- Is the OPC heterogeneity, i.e. are all OPCs capable of doing this or is it only some? Along this line, in Fig3, a quantification of how many OPCs (%) express AXL
- Fig3, the overlaid staining looks quite messy and the shape of OPCs cannot really be determined
- In Fig 2b, the zoomed-in cell is not the same as in the overview image
- Fig 2d, please show the single channel for PDGFRa, the small overlay is difficult to judge
- Fig1: the authors claim to see MBP structures that resemble myelin, however, from the images this does not become evident, also the large overview images are blurry, but it looks like there are only very branched MBP-positive cells
- Fig 1, in some images the DIV (e.g in the Iba1 staining) is missing
- Why is the UNC2025 not used in the organoid setup as well, to prove that it really works?
- Figure 4: whilst the IHC and quality of the organoid system looks really good, the quality of the pure OPC cultures needs to be questioned based on the data provided. OLIG2 is a nuclear marker, but the staining in the image seems to be unspecific, also the shape of the PDGFRa cells is quite strange and everything seems to be stained. Similarly, in previous figures, the authors very nicely provided 3D-reconstructions to see that something is intracellular, however here, the pHrodo-Dye seems unspecific and it is hard to tell whether it is intracellular or not (4d)

(Remarks on code availability)

Reviewer #4

(Remarks to the Author)

(Remarks on code availability)

I did not review the code. The github URL sent me to a 404 error page when I tried to access it.

Version 1:

Reviewer comments:

Reviewer #1

(Remarks to the Author)

In this revised manuscript the authors have impressively addressed all of my comments and have added significant new data. Hence the manuscript is much more complete now. It clearly demonstrates that a subset of human OPCs in organoid culture internalizes synaptic material via GAS6-TAMRTK interaction via the TAM-RTK AXL expressed on the OPCs. The manuscript is solid, provides a novel mechanistic insight into the mechanism of synapse internalization by OPCs and represents an important new concept that will be of interest to the broad neuroscience community.

(Remarks on code availability)

Reviewer #2

(Remarks to the Author)

I feel the authors have done a nice job in responding to most of my initial concerns, as well many of those raised by the other reviewers. Overall, it is my opinion that the manuscript is suitable for publication.

(Remarks on code availability)

Reviewer #3

(Remarks to the Author)

The authors have put a lot of work into the revision of this manuscript, which has highly improved the quality, especially in the figures. They have addressed most of my previously raised concerns and added new experiments. However, in Fig4 a and j, I am still not convinced of the staining, the OLIG2 in a seems all over the place and SOX10 in j does not seem to be nuclear. If the images look like this, how were they able to quantify positive cells?

(Remarks on code availability)

Reviewer #4

(Remarks to the Author)

(Remarks on code availability)

Code is now accessible and appears reasonable and appropriate.

Version 2:

Reviewer comments:

Reviewer #3

(Remarks to the Author)

I am very happy with the further changes and the representative images look a lot clearer. And I think the manuscript is now suitable for publication.

(Remarks on code availability)

made.

Point-by-point response to reviewers' comments

We thank the reviewers for their valuable feedback, which has substantially strengthened the study. In particular, we have generated new organoids, as well as a co-culture system, to confirm the effect of UNC2025 on AXL+ OPCs. Below, we address each comment in detail and have highlighted all changes made in the manuscript in yellow.

Reviewer #1 (Remarks to the Author):

In this manuscript, the authors describe the role of GAS6-AXL signaling in promoting phagocytosis of neuronal synaptic terminals using a multi-lineage organoid system that includes microglia and OPCs generated from hiPSCs. They replicate the observations made by others using the rodent system and show that OPCs phagocytose synaptic material. They then performed an analysis of inferred cell-cell communication networks from snRNA-seq and identified GAS6-AXL as the ligand receptor pair likely to be involved in this process. This was verified by showing the presence of AXL mRNA levels in OPC monocultures and by immunodetection of GAS6 deposited on OPCs (mRNA made by neurons and microglia). The levels of AXL and GAS6 in OPCs correlated with the number of synaptic puncta internalized. To further establish the causal relationship, the authors inhibited GAS6-AXL signaling in OPC monocultures by a small molecule AXL inhibitor or AXL siRNA and showed that these manipulations reduced phagocytosis of human synaptosomes, as shown by a reduction in pHrodo+ phagocytotic index.

This is a timely study providing a novel mechanistic insight into the signaling pathways that are involved when OPCs internalize neuronal synaptic terminals. The manuscript is put together very tightly in four sets of figures, each containing to significant new data and

organized in a lean and logical manner. The authors used an elegant multi-lineage organoid system to include both OPCs and microglia, as previous rodent studies had shown the requirement of microglia in OPC-mediated phagocytosis, though the authors did not show the role of microglia in their current human organoid culture model other than to show that the total internalized GEPH puncta / cell was similar between OPC and microglia.

We thank the Reviewer for the thorough evaluation of our manuscript and the constructive feedback that has significantly improved the study.

I have a few minor points that could be addressed for improved clarification.

1) The results section describing spontaneous uptake of synaptic structures by OPCs (Fig. 2) seems somewhat truncated and cryptic. While I realize that there is a page/word count limit, this part could be explained in more depth, as it describes the assay that forms the basis for all the key findings in the manuscript. The authors provide images for PSD-95 and SYN1 but perform 3D volumetric reconstruction using GEPH. The authors should explain their unique culture model and the cellular composition in more detail. Below are some specific points that should be addressed.

a) What was the abundance of excitatory and inhibitory neurons? (They mention GE patterning but also show excitatory synapses).

Consistent with ganglionic eminence patterning through SHH pathway activation and WNT inhibition, the neuronal clusters at DIV250 comprised approximately 25% excitatory neurons, 71% inhibitory neurons, and 4% unclassified neurons. We have now added this in the revised manuscript as the new **Supplementary Fig. 2h**. The corresponding text in the revised **Results** section reads as follows (lines 496-501):

To quantify neuronal composition within the organoids, we scored neuronal clusters using curated gene signatures derived from human fetal cortical development¹, which classified 71.2% of neurons as inhibitory and 24.8% as excitatory, with the remaining 4.0% unclassified (Supplementary Fig. 2h). These proportions reflect the expected bias toward ventral forebrain identities under GE patterning conditions, as also observed qualitatively by positive immunolabelling of vGLUT1 and GABA (Fig. 1c).

The corresponding text in the revised **Methods** section reads as follows (lines 298-304):

To quantify the excitatory and inhibitory composition of neuronal clusters, cells within neuronal clusters were scored using gene signatures obtained from fetal cortical neurons as described by Velmeshev *et al.*, 2023 (signature pattern = "burst").¹ Signature scores were computed for excitatory and inhibitory lineages, and a delta threshold of 0.01 was applied to assign each cell as excitatory, inhibitory, or unclassified. Cell-type proportions were subsequently computed across all neuronal clusters to assess neuronal subtype composition within the organoid model.

b) How are microglia involved in OPC-mediated phagocytosis of synaptic terminals? For example,

i) Are microglia always found near the sites of internalization of synapses by OPCs (i.e. at AXL+ sites?)

We also observe synaptic uptake in OPCs without microglia in close proximity at the time of quantification (**Fig. 4I**).

ii) Have the authors tried removing microglia from the multi-lineage organoids?

We have now examined the uptake of synaptic material in ventral forebrain-patterned organoids lacking microglia (**Methods**, lines 152-173). Consistent with previous animal

studies,² which reported approximately a 25% reduction in OPC uptake following microglia depletion with PLX5622, our results show that OPCs retain the capacity to internalize synaptic material even in the absence of microglia (**Fig. 4l-m**).

We now include this data in the **Results (Fig. 4l-m)** and discuss the relevance in the revised **Discussion** section (lines 622-624):

This aligns with animal studies showing reduced ability of OPCs to engulf thalamocortical inputs following depletion of microglia, one of the two major sources of AXL-activating GAS6, though residual uptake persists, potentially mediated by neuronal-derived GAS6.² Consistent with this, we found that, even in the absence of microglia, OPCs retained the capacity to internalize synaptic elements, underscoring their intrinsic ability to engage in synaptic remodelling.

iii) If pHrodo signal increases in OPC monocultures in the absence of microglia (Fig. 4), how is this finding interpreted? Are microglia not necessary for phagocytosis of synapses by OPCs? Are there a sufficient number of microglia among the 35% of the non-OPC cells in the OPC monocultures? Or does exposure to synaptosomes trigger a different signaling pathway?

We have examined the cell composition of the OPC monocultures by immunocytochemistry and found no detectable microglia among the non-OPC cells (**Supplementary Fig. 3i**). This is in line with the differentiation protocol, which does not include PMPs, or other mesoderm-lineage cells and the conditions required for microglial generation. Notably, GAS6, the AXL ligand, is also expressed by neurons, and we confirmed that GAS6 protein is present on neuron-derived synaptosomes (**Fig. 4b**). This suggests that the GAS6-AXL pathway can also

be activated without microglia, although microglia-produced GAS6 may promote further activation, explaining the decrease in OPC uptake after microglia depletion in other studies.²

We now provide additional clarification in the **Results** section of the revised manuscript:

(Lines 559-567): To confirm a mechanistic role of TAM-RTK activation in OPC-mediated synapse elimination, we then derived OPCs from iPSCs ($n=3$ subjects) in mono-cultures (Fig. 4a, Supplementary Fig. 3h) and exposed them to human synaptosomes derived from iPSC-derived neurons, labeled with a pH-sensitive dye (pHrodo). Immunocytochemistry confirmed that these cultures were devoid of microglia (Supplementary Fig. 3i), consistent with the absence of mesodermal progenitors in the differentiation protocol. Although these OPC monocultures lack both neurons and microglia, immunostaining showed that synaptosomes – in addition to the synaptic markers (SYN1 and GEPH) – contained detectable levels of GAS6 (Fig. 4b), the ligand for AXL, suggesting that neuronal GAS6 on synaptosomes may be sufficient to engage AXL on OPCs.

(Lines 595-600): Finally, we also used UNC2025 in the organoid model. To avoid effects modulated by inhibition of AXL on microglia, we utilized ventral forebrain-patterned organoids lacking PMPs. First, we confirmed that OPCs retained the ability to internalize synaptic material also in absence of microglia and then observed a marked reduction in internalized synapses by AXL⁺ OPCs in inhibitor- compared to vehicle-treated organoids (Fig. 4l-m), supporting the conclusion that AXL signalling promotes synaptic engulfment by OPCs.

2) Of the GAS6 ligands, TYRO3 might also be expressed in oligodendrocyte lineage cells (Zhang et al., 2014; Marques et al., 2018). Can GAS6 bind to TYRO3, and would GAS6-TYRO3 interaction influence/modulate OPC-mediated synapse uptake?

We appreciate the Reviewer's comment and we agree that TYRO3 can also be expressed in oligodendrocyte-lineage cells. While GAS6 is indeed a ligand for all three TAM receptors (TYRO3, AXL, and MERTK), it exhibits the highest binding affinity and activation potency for AXL – approximately 100-fold higher than for TYRO3.^{3,4} To investigate the relative contribution of these receptors, we used the small molecule inhibitor UNC2025 at variable concentrations: at 150 nM, UNC2025 selectively inhibits AXL, while at 200 nM it additionally inhibits TYRO3.⁵ In our assay, inhibition of synapse uptake in OPCs was comparable at both concentrations (**Fig. 4g**), indicating that further inhibition of TYRO3 does not augment the effect. This suggests AXL as a more prominent mediator of synapse uptake in our system, verified by siRNA KD (**Fig. 4i**).

3) To what extent is GAS6 colocalized with AXL on OPC membranes? i.e. Could the authors show co-localization of GAS6 deposited by neuron/microglia and AXL endogenously produced by OPCs? Does the extent of GAS6-AXL colocalization along the the OPC surface correlate with the extent of synapse phagocytosis, perhaps to an even greater extent than the correlation with the level of AXL?

We thank the Reviewer for the insightful question regarding GAS6 and AXL co-localization on OPCs. We agree that demonstrating the spatial relationship between GAS6, which is deposited by neurons and microglia, and AXL, which is endogenously expressed by OPCs, is critical to strengthen the mechanistic link in our model. Due to technical limitations with antibody species and available fluorophores, we were unable to simultaneously co-stain for PDGFR α , AXL, GAS6, and a synaptic marker in the same sample. However, we have now included high-resolution confocal images and volumetric reconstruction showing clear co-localization of GAS6 and AXL on PDGFR α + OPCs (see revised **Fig. 3c**), supporting the presence of ligand-receptor interactions at the OPC surface. In addition, new supplementary

data (**Supplementary Fig. 3e**, and lines 547-548) show that OPCs lacking detectable AXL expression also lack GAS6 labeling, whereas AXL+ OPCs display robust GAS6 signal, supporting that GAS6 deposition occurs preferentially at AXL+ OPC membranes, potentially facilitating synaptic uptake.

4) The data in Fig. 4 show that AXL is involved in phagocytosis of exogenous human synaptosomes. Since these ‘OPC monocultures’ do not contain neurons or microglia, how is AXL activated in this culture system? Does the synaptosome prep contain GAS6 or does that bypass the need to have GAS6 activate TAM-RTKs on OPCs and somehow activates AXL on OPCs by a separate mechanism? How do the levels of AXL in OPC monocultures compare with those in the organoids that contain microglia and neurons?

We thank the Reviewer for these important questions. As described above, we have now confirmed by immunostaining that the neuron-derived synaptosomes are positive for both SYN1 and GAS6 (**Fig. 4b**), suggesting that synaptosome-associated GAS6 may engage AXL receptors on OPCs in the absence of microglia. Regarding AXL expression levels, we used qPCR to assess *AXL* transcript levels in OPC monocultures (**Fig. 4f**) and immunohistochemistry to assess AXL protein localization (**Fig. 3c-d**) and abundance (**Supplementary Fig. 3c**) in organoids. Due to these methodological differences, a direct quantitative comparison is limited, but both approaches show that AXL is robustly expressed by OPCs across our culture systems.

We now provide additional clarification in the **Results** section of the revised manuscript (Lines 559-567):

To confirm a mechanistic role of TAM-RTK activation in OPC-mediated synapse elimination, we then derived OPCs from iPSCs ($n=3$ subjects) in monocultures (Fig. 4a, Supplementary Fig. 3i) and exposed them to human, pHrodo-labeled synaptosomes derived

from iPSC-derived neurons. Immunocytochemistry confirmed that these cultures were devoid of microglia (Supplementary Fig. 3i), consistent with the absence of mesodermal progenitors in the differentiation protocol. Although these OPC monocultures lack both neurons and microglia, immunostaining showed that synaptosomes contained synaptic markers (SYN1 and GEPH) and detectable levels of GAS6 (Fig. 4b), the ligand for AXL, suggesting that neuronal GAS6 on synaptosomes may be sufficient to engage AXL on OPCs.

Overall, this is a well written manuscript that effectively communicates exciting new data on the signaling pathway that mediates phagocytosis of neuronal synapses by OPCs in a human organoid culture system.

Reviewer #1 (Remarks on code availability):

Looks OK but I am not an expert in single nuclei RNA-seq bioinformatics.

Reviewer #2 (Remarks to the Author):

In this report, Gkogka et al aim to better characterize how oligodendrocyte precursor cells (OPCs) and microglia contribute to synaptic development, namely, by utilizing induced pluripotent stem cell (iPSC) and organoid based models. The authors employ use of IHC, scRNA-seq, and functional assays to identify that OPCs and microglia internalize synaptic elements within organoid models. Even further, the authors utilize ligand-receptor analysis tools to implicate neuronal and microglial GAS6-mediated activation of OPC internalization of synaptic elements. Live-imaging functional assays of engulfment using monocultured OPCs further demonstrate that inhibition of the GAS-TAM signaling pathway reduces OPC internalization of synaptic elements. Altogether, the authors' work helps further understanding of signaling events relevant to synaptic development, health, and maintenance within a novel, reductionist model. These findings also establish some foundation for future

work to evaluate the GAS-TAM pathway as it relates to neurodevelopment, aging, and neurodegeneration. Some major notes are identified that relate to the authors' generalization of findings that require attention, with particular emphasis on better characterization and benchmarking of the oligodendrocyte lineage in their *in vitro* culture system. Minor notes relate to clarifying small sections that contain suspected error.

We would like to thank Reviewer #2 for the thoughtful and constructive comments. Below we address each point raised, incorporating the suggestions to the revised manuscript.

Major

1. The multilineage model is intriguing, but there are VERY few microglia (Fig 1b, 1g) and so it is important I think to be much more outward about the fact that the myeloid compartment is not truly reflective of *in vivo* proportions.

We fully agree with the Reviewer that brain organoid models with integrated myeloid cells still have clear limitations and do not fully recapitulate the *in vivo* myeloid compartment. The estimated proportion of IBA1⁺ cells in our system was low (~1-3%) consistent with other studies using brain organoid models with integrated myeloid cells, such as Ormel *et al.*,⁶ Cakir *et al.*,⁷ and Muffat *et al.*,⁸ which report similar microglial abundance in the range of ~1-5%. We have updated the Results section to clarify this accordingly.

The corresponding revised **Results** section reads as follows (lines 491-496):

Notably, while the abundance of microglia was relatively low, a feature consistent with other brain organoid models^{19,43,44}, and not fully representative of *in vivo* microglial proportions, the microglia that emerged within the organoid model exhibited high transcriptional similarity to primary fetal microglia from the developing human forebrain and displayed a core microglial signature³⁴ (Supplementary Fig. 2g).

2. I have a general concern about the use of PDGFR α (and even Olig2) in forebrain organoids to specifically mark OPCs. In this model and at these developmental stages, PDGFR α can also mark glial progenitors (not necessarily OPC-specific). Therefore, a lot of the conclusions that depend on the PDGFR α lineage may mistakenly be attributed to OPCs. I think additional stains like SOX10 would be much more convincing that cells of interest (e.g. Fig 3C) are truly in an oligo-committed lineage.

We appreciate the Reviewer's concern regarding the specificity of PDGFR α as a marker for OPCs at early developmental stages. To directly address this, we performed additional immunostainings for SOX10, a well-established oligodendrocyte-lineage marker, and quantified the overlap between PDGFR α ⁺ cells and PDGFR α ⁺SOX10⁺ double-positive cells across 3 independent lines. The analysis revealed that 97% of PDGFR α ⁺ cells also express SOX10, supporting their identity as oligodendrocyte-lineage cells rather than more general glial progenitors. Based on this high degree of overlap, we used PDGFR α ⁺ cells as a robust proxy for OPCs in our correlation analyses of AXL and GAS6 protein levels with synaptic uptake. For the Reviewer's reference, we provide the quantification below (**Figure R1**) comparing the proportion of PDGFR α ⁺ versus SOX10⁺PDGFR α ⁺ double-positive OPCs, visually confirming the high degree of marker overlap. As this experiment specifically addresses the Reviewer's concern, we have included these data here as supplementary validation rather than in the main manuscript figures.

Figure R1: Quantification of SOX10⁺PDGFR α ⁺ double-positive cells out of total PDGFR α ⁺ cells across 3 independent lines. Data are mean \pm s.e.m.; $n = 3$ lines; each point represents the average per organoid.

3. The single nucleus data in the UMAP in 1g is a bit confusing. At day 250 it is surprising that the OL-lineage appears to remain so immature in these cultures. Along these lines, I can understand the comparison to Marton et al., but I'd instead recommend benchmarking the oligos in this model with a primary fetal dataset and not another organoid dataset exclusively. Altogether, this analysis could be performed more robustly and could be presented more clearly to indicate the degree of OL-lineage differentiation present in this model. If it's true that the oligodendrocytes do not fully mature by day 250, this should also be discussed more clearly.

We thank the reviewer for raising this point and the helpful suggestion. In response, we re-analyzed the OL-lineage cells by mapping them to a comprehensive primary fetal brain reference dataset encompassing both prenatal and postnatal stages.¹ This approach provides a more robust evaluation of oligodendrocyte differentiation in our model. Notably, OL-lineage cells predominantly aligned with second-trimester profiles, with a subset showing transcriptional similarity to more mature, postnatally captured oligodendrocytes (**Fig. 1j**).

We have updated the **Results** section accordingly (lines 488-491):

Importantly, the OL cell cluster consisted of a spectrum of OL-lineage cells, including proliferating OPCs, OPC populations and **relatively mature OLs (Fig. 1i), displaying high transcriptional similarity with primary second-trimester OL cells (Fig. 1j)**, as well as with other organoid OPC populations (Supplementary Fig.2e-f).

4. Lines 416-427, Fig. 2: GEPH is used as a marker to measure microglial and OPC phagocytic capacity. GEPH is typically associated with inhibitory elements, and my concern is, that without additional markers to measure phagocytosis, the authors' conclusions may not be easily generalized all phagocytic functions. Can the authors provide data for any additional markers, namely for excitatory postsynaptic elements (PSD-95, for example)?

We agree with the Reviewer that using GEPH alone, an inhibitory synaptic marker, may limit generalizability to all types of synaptic elements. We selected GEPH for quantification of synapse uptake because the neuronal population in our organoids is predominantly inhibitory (71.2% inhibitory, 24.8% excitatory; Supplementary Fig. 2h), making it the most representative marker for our model. However, in response to the Reviewer's request, we now provide qualitative evidence that OPCs can also internalise excitatory synaptic elements. Using PSD-95 immunostaining followed by 3D volumetric reconstruction, we confirm internalization of excitatory postsynaptic material by OPCs in both organoid and monoculture contexts. These new images are included in **Fig. 2c** and **Fig. 4e**.

The revised **Results** section reads as follows (lines 506-514):

High-resolution confocal imaging at 130 DIV revealed that both microglia and OPCs made contacts with neuronal synapses of glutamatergic (PSD-95+) and GABAergic (GEPH+) identities (Fig. 2a-e). Three-dimensional volumetric reconstruction revealed OPCs demonstrating synaptic engulfment in varying degrees of internalization, ranging from surface association to fully phagocytosed post-synaptic terminals within phagolysosomal compartments, similar to microglia, (Fig. 2d-e). Given that the organoids were strongly enriched in inhibitory neurons under our GE patterning conditions (Supplementary Fig. 2h, Fig. 1c), quantitative analyses focused on GEPH+ post-synaptic terminals as the predominant synapse type.

5. Lines 462-86, Fig. 4: Similar to Major note (4.), the functional experiments use PSD-95 as a marker for phagocytosis, yet lack additional data on GEPH markers which were indicative of function in previous experiments. Altogether, consistency in markers and methods would benefit these works in order to strengthen the generalization of these findings.

We agree and have now included additional staining for GEPH in synaptosome preparations, alongside SYN1 (**Fig. 4b**), to confirm the presence of both inhibitory and excitatory synaptic elements in the particles internalized by OPCs.

Minor

1. Line 200: Specify “couple of hours” to a standard numeric value.

Corrected to: “for 2 h” (line 228).

2. Lines 447-448: Reference to Fig. 3b-d seems mismatched, perhaps this is meant to be Fig. 3a-b or S3a-b?

We thank the Reviewer for noticing this. The reference has been corrected to “**Fig. 3a-b**” (line 538).

3. Lines 447-450: Difficult to read through and would benefit from clarifying separation between the authors findings/models with that of others. See Minor note (2.) For additional notes of confusion for this section.

We agree with the Reviewer’s feedback, and we revised this **Results** section to more clearly separate our findings from those in previous studies, as follows (lines 537-542):

In line with our inferred neuron-microglia signaling network in the forebrain organoids (**Fig. 3a–b**), previous *in vivo* studies in mice have shown that TAM receptors play key roles in synapse and cell debris clearance. Specifically, MERTK and AXL deficiency leads to reduced microglial engulfment of apoptotic cells during adult neurogenesis,^{10,11} and MERTK has been implicated in astrocyte-mediated synapse elimination during visual system development.¹²

4. Line 462: Consider changing heading based upon experimental design. AXL was not activated in any way; it was found that increased inhibition of AXL reduces internalization by

OPCs. Perhaps something like “AXL inhibition impairs uptake of synaptic material in OPCs” to better reflect the experimental design and results. Otherwise, I would also recommend overactivation of TAM signaling via GAS6 supplementation, microglial/neuronal conditioned media, or genetic manipulations to further evaluate the question of TAM activation by GAS6.

We thank the reviewer for this helpful suggestion. We agree that the previous heading could be misinterpreted, thus, to better reflect the experimental design and the primary finding – namely that inhibition of AXL impairs synaptosome uptake – we have revised the section title to: “AXL inhibition impairs uptake of synaptic material in OPCs” (line 558).

5. I don't believe the velocity in the UMAP in 1g is necessary – if anything it is a bit distracting from interpreting the clusters.

We agree that the RNA velocity overlay may distract from interpretation. We have removed RNA velocity from **Fig. 1g** in the revised manuscript and simplified the visualization to better highlight the lineage clusters.

Reviewer #3 (Remarks to the Author):

In their manuscript entitled “Human oligodendrocyte progenitor cells mediate synapse elimination through TAM receptor activation” Gkogka et al. use a human iPSC-derived organoid system to assess the role of oligodendrocyte progenitor cells (OPCs) in synapse pruning. The authors suggest that similar to microglia, OPCs engulf synapses during development, which seems to be mediated via the AXL receptor – a phagocytosis receptor that is well known for playing an important role in microglia.

These results are quite interesting and suggest a yet unknown function of OPCs and is this of

high importance for the field. Although the paper is quite good as a beginning, there are still fundamental experiments lacking before it can be published.

- Is there evidence that the expression of AXL is not only an in-vitro artefact? Some human (published data on this). Could it be that it is only transient during development (as represented by the organoid model)? This suggestion is a yet unknown function of OPCs and thus of really important. Can this be seen in mouse models during different stages of development and in the adult brain? Or is it human-specific?

We thank the reviewer for this important question. There is evidence that AXL expression is not merely an *in vitro* artifact. Multiple single-nucleus RNA-seq datasets report *AXL* expression in different OPC subpopulations. For example, in the Nowakowski *et al.* single-nucleus RNA-seq dataset (Science, 2017)¹³ 14% of the sampled OPCs in the human fetal cortex expressed AXL, a proportion closely matching our findings (16%, **Supplementary Fig. 3c**) in the organoid model.

Regarding expression across the lifespan, developmental mouse datasets and adult human brain tissue show markedly lower AXL transcript levels (often below 1%). However, this trend of low mRNA *AXL* levels in these datasets also applies to microglia despite extensive evidence from protein-based and functional studies demonstrating key roles for AXL in microglial biology during these periods. As this discrepancy suggests technical limitations, including transcript dropouts in single-nucleus RNA-seq and context-dependent regulation of AXL expression, we believe it is premature to conclude that its expression is strictly transient during certain developmental periods, although we agree with the reviewer that this warrants further studies (line 648-651). We have now added in the revised **Results** section of the manuscript (lines 544-546):

We found that 16% of OPCs harbored AXL protein (Fig. 3c, Supplementary Fig. 3c), closely matching the 14% reported in primary fetal cortical tissue,¹³ while MERTK protein (Supplementary Fig. 3d).

- Is there OPC heterogeneity, i.e. are all OPCs capable of doing this or is it only some? Along this line, in Fig3, a quantification of how many OPCs (%) express AXL.

This is an important aspect. To address this, we analyzed OPCs in our organoid model across three independent lines, quantifying AXL expression within OPC population, and found that approximately 16.4% of PDGFR α +SOX10+ double-positive cells express detectable levels of AXL (lines 544-546, Supplementary **Fig. 3c**). Using Imaris-based 3D volumetric reconstructions, we also visualized OPCs at varying stages of synapse engulfment, ranging from synaptic particles localized only to the cell surface to extensive internalization (**Fig. 2d-e**). These data indicate heterogeneity in both AXL expression and synaptic uptake among OPCs, consistent with emerging literature describing OPCs as a heterogeneous population with multiple functional states, of which AXL expression may mark one phenotype associated with phagocytic activity.

The revised **Results** section now reads as follows (lines 508-511):

Three-dimensional volumetric reconstruction revealed OPCs demonstrating synaptic engulfment in varying degrees of internalization, ranging from surface association to fully phagocytosed post-synaptic terminals within phagolysosomal compartments, similar to microglia, (Fig. 2d-e).

- Fig3, the overlaid staining looks quite messy and the shape of OPCs cannot really be determined.

We thank the reviewer for this comment and apologize for the unclarity. We have updated the panels in **Fig. 3d-f** to include improved high-resolution images and Imaris-based 3D rendering with clearer outlines of OPCs and reduced channel crowding to highlight the intracellular localization of synaptic markers.

- In Fig 2b, the zoomed-in cell is not the same as in the overview image.

We thank the reviewer for pointing this out. It has now been corrected (**Fig. 2b**).

- Fig 2d, please show the single channel for PDGFR α , the small overlay is difficult to judge.

We have updated **Fig. 2d-e** to include individual channels for PDGFR α and IBA1, alongside DAPI indicating nuclei, to enable clearer distinction of the OPC and microglial compartments.

- Fig1: the authors claim to see MBP structures that resemble myelin, however, from the images this does not become evident, also the large overview images are blurry, but it looks like there are only very branched MBP-positive cells.

We acknowledge this limitation. To clarify this point, we now include higher-resolution zoom-ins showing MBP wrapping around MAP2⁺ neurites, with volumetric reconstruction, suggestive of ensheathment (**Fig. 1d**). We also clarify in the text that while MBP⁺ processes are present, compact myelin was not observed.

The revised **Results** section of the manuscript now reads as follows (lines 469-471):

From 43 DIV to 130 DIV, a substantial fraction of OPCs robustly differentiated into OLs with MBP⁺ processes frequently observed in proximity to neurites, yet compact myelin was not observed (Fig. 1d).

- Fig 1, in some images the DIV (e.g in the Iba1 staining) is missing.

We thank the reviewer for pointing this out. We apologize for the oversight and have now added the missing DIV information to all relevant figure panels in **Fig. 1** to ensure clarity.

- Why is the UNC2025 not used in the organoid setup as well, to prove that it really works?

We agree with the reviewer and have now performed experiments with UNC2025 in the organoid model. To avoid effects modulated by inhibition of AXL on microglia, we used ventral forebrain organoids containing OPCs but not microglia. Exposure to 150 nM UNC2025 (the concentration that inhibits AXL) for one week resulted in a significant reduction in synapse internalization by AXL+ OPCs, as quantified by immunostaining, confocal microscopy, and Imaris analysis (**Fig. 4l-m**).

Given cellular complexity of the forebrain organoids, we also performed experiments in a more reductionist model consisting of OPC-neuron co-cultures (**Fig. 4j**). These co-cultures were treated with 150 nM UNC2025 or vehicle control from day 22 to day 26 and assessed for OPC-mediated synapse uptake via immunohistochemistry, confocal imaging, and 3D reconstruction using Imaris. Again, we observed a consistent and statistically significant reduction in synaptic internalization by OPCs in UNC-treated cultures (**Fig. 4k**), confirming the inhibitor's effect also in the neuron-OPC co-culture environment.

We provide the methodology for the additional culture models in the revised Methods section (lines 152-173, 416-443) and the additional data in the **Results** section of the revised manuscript (lines 588-600):

To assess the functional role of AXL in synapse uptake by OPCs in a more physiologically relevant context, we next established co-cultures of iPSC-derived OPCs and neurons (Fig. 4j) and treated the cultures with 150 nM UNC2025 or vehicle control between days 24-28.

Immunostaining followed by 3D reconstruction revealed a significant reduction in the uptake of SYN1+ material by PDGFR α + OPCs upon UNC2025 treatment (Fig. 4k), confirming the role of AXL and TAM-RTK signalling in OPC-mediated synaptic uptake under co-culture conditions.

Finally, we also used UNC2025 in the organoid model but then utilized ventral forebrain-patterned organoids lacking PMPs to avoid effects modulated by inhibition of AXL on microglia. First, we confirmed that OPCs retained the ability to internalize synaptic material also in absence of microglia and then observed a marked reduction in internalized synapses by OPCs in inhibitor- compared to vehicle-treated organoids (Fig. 4l-m), supporting the conclusion that AXL signalling promotes synaptic engulfment by OPCs.

- Fig. 4: whilst the IHC and quality of the organoid system looks really good, the quality of the pure OPC cultures needs to be questioned based on the data provided. OLIG2 is a nuclear marker, but the staining in the image seems to be unspecific, also the shape of the PDGFR α cells is quite strange and everything seems to be stained. Similarly, in previous Fig.s, the authors very nicely provided 3D-reconstructions to see that something is intracellular, however here, the pHrodo-Dye seems unspecific and it is hard to tell whether it is intracellular or not (4d).

We appreciate the Reviewer's comment. We have now re-imaged OPC cultures with improved OLIG2 and PDGFR α staining (Fig. 4a), and performed 3D Imaris analysis of pHrodo+ synaptosomes co-localized with OLIG2+PDGFR α + OPCs, confirming intracellular localization (Fig. 4e).

Reviewer #4 (Remarks to the Author):

Reviewer #4 (Remarks on code availability):

I did not review the code. The github URL sent me to a 404 error page when I tried to access it.

We thank Reviewer #4 for their time and efforts in co-reviewing our manuscript as part of the Nature Communications initiative. We apologize for the inconvenience and provide the functional repository link <https://github.com/SellgrenLab/organoid-oligodendrocyte>, which has been updated in the manuscript (lines 451-453) and supplementary materials to ensure access to all code and analysis scripts.

References

1. Velmeshev, D. *et al.* Single-cell analysis of prenatal and postnatal human cortical development. *Science (1979)* **382**, (2023).
2. Auguste, Y. S. S. *et al.* Oligodendrocyte precursor cells engulf synapses during circuit remodeling in mice. *Nature Neuroscience* 2022 25:10 **25**, 1273–1278 (2022).
3. Lemke, G. & Rothlin, C. V. Immunobiology of the TAM receptors. *Nature Reviews Immunology* 2008 8:5 **8**, 327–336 (2008).
4. Lew, E. D. *et al.* Differential TAM receptor-ligand-phospholipid interactions delimit differential TAM bioactivities. *Elife* **3**, (2014).
5. Zhang, W. *et al.* UNC2025, a potent and orally bioavailable MER/FLT3 dual inhibitor. *J Med Chem* **57**, 7031–7041 (2014).
6. Ormel, P. R. *et al.* Microglia innately develop within cerebral organoids. *Nature Communications* 2018 9:1 **9**, 1–14 (2018).
7. Cakir, B. *et al.* Expression of the transcription factor PU.1 induces the generation of microglia-like cells in human cortical organoids. *Nature Communications* 2022 13:1 **13**, 1–15 (2022).
8. Muffat, J. *et al.* Efficient derivation of microglia-like cells from human pluripotent stem cells. *Nat Med* **22**, 1358 (2016).
9. Nowakowski, T. J. *et al.* Spatiotemporal gene expression trajectories reveal developmental hierarchies of the human cortex. *Science* **358**, 1318–1323 (2017).
10. Furgeaud, L. *et al.* TAM receptors regulate multiple features of microglial physiology. *Nature* **532**, 240 (2016).
11. Diaz-Aparicio, I. *et al.* Microglia Actively Remodel Adult Hippocampal Neurogenesis through the Phagocytosis Secretome. *The Journal of Neuroscience* **40**, 1453 (2020).
12. Chung, W. S. *et al.* Astrocytes mediate synapse elimination through MEGF10 and MERTK pathways. *Nature* **504**, 394–400 (2013).
13. Nowakowski, T. J. *et al.* Spatiotemporal gene expression trajectories reveal developmental hierarchies of the human cortex. *Science (1979)* **358**, 1318–1323 (2017).

Point-by-point response to reviewers' comments (Revision #2)

We thank all reviewers for their constructive feedback and for recognizing the improvements in the revised manuscript. Below we address the remaining points raised.

Reviewer #3

The authors have put a lot of work into the revision of this manuscript, which has highly improved the quality, especially in the figures. They have addressed most of my previously raised concerns and added new experiments.

We thank the Reviewer for this positive assessment.

However, in Fig4 a and j, I am still not convinced of the staining, the OLIG2 in a seems all over the place and SOX10 in j does not seem to be nuclear. If the images look like this, how were they able to quantify positive cells?

We have now updated Fig. 4a to more clearly demonstrate the nuclear localization of both SOX10 and OLIG2. For Fig. 4j, we replaced the previous panel with a higher-magnification image that distinctly highlights nuclear SOX10. Because the OPCs in the co-cultures were generated using the same protocol as in monocultures, we have moved this new representative image to the Supplementary Information (Supplementary Fig. 3I). In Fig. 4j, we now instead provide representative Imaris 3D reconstructions that were directly used in our quantitative analyses for each condition, thereby clarifying how the data were obtained.